# Young KRAB-zinc finger gene clusters are highly dynamic incubators of ERV-driven genetic heterogeneity in mice

Melania Bruno [1] ✉, Sharaf M. Farhana[1], Apratim Mitra [1], Kevin Costello[2], Dawn E. Watkins-Chow[1], Glennis A. Logsdon [3], Craig W. Gambogi[4], Beth L. Dumont [5], Ben E. Black [4], Thomas M. Keane [6], Anne C. Ferguson-Smith [2], Ryan K. Dale [1] & Todd S. Macfarlan [1] ✉

KRAB-zinc finger proteins (KZFPs) comprise the largest family of mammalian transcription factors, rapidly evolving within and between species. Most KZFPs in human and mice have been found to repress endogenous retroviruses (ERVs) and other retrotransposons, with KZFP gene numbers correlating with the ERV load across species, suggesting coevolution. Whether new KZFPs emerge in response to ERV invasions is currently unknown. Using a combination of long-read sequencing technologies and genome assembly, we present a detailed comparative analysis of young KZFP gene clusters in the mouse lineage, which has undergone recent KZFP gene expansion and ERV infiltration. Detailed annotation of KZFP genes in a cluster on *Mus musculus* Chromosome 4 reveals parallel expansion and diversification of this locus in different mouse strains (C57BL/6 J, 129S1/SvImJ and CAST/EiJ) and species (*Mus spretus* and *Mus pahari*). Our data supports a model by which new ERV integrations within young KZFP gene clusters likely promoted recombination events leading to the emergence of new KZFPs that repress them. At the same time, ERVs also increased their numbers by duplication instead of retrotransposition alone, unraveling a new mechanism for ERV enrichment at these loci.

KRAB-zinc finger proteins (KZFPs) comprise a large family of transcription factors, numbering in the hundreds in most mammalian genomes[1]. KZFPs are characterized by a variable array of tandem C2H2 zinc fingers conferring DNA binding specificity, and a Krüppel-associated box (KRAB) domain recruiting the SUMO ligase KAP1/TRIM28, that engages with several heterochromatin forming complexes to induce gene silencing[2]. While a few evolutionary old and conserved KZFPs have been shown to play essential roles in core developmental processes like embryonic development[3], imprinting[4,5] and meiotic hotspot determination[6,7], the vast majority of KZFPs, both young and old, bind to and repress transposable elements (TEs)[1]. In particular, several evolutionary young and clade or species specific KZFPs have been shown to silence endogenous retroviruses (ERVs) of similar age[8,9]. This finding, together with a striking correlation between the number of zinc finger genes and the load of ERVs in different mammalian genomes, has led to a model of KZFP and ERV

[1]The Eunice Kennedy Shriver National Institute of Child Health and Human Development, National Institutes of Health, Bethesda, MD, USA. [2]Department of Genetics, University of Cambridge, Downing Street, Cambridge, UK. [3]Department of Genetics, Epigenetics Institute, Perelman School of Medicine, University of Pennsylvania, Philadelphia, PA, USA. [4]Department of Biochemistry & Biophysics, Penn Center for Genome Integrity, Epigenetics Institute, Perelman School of Medicine, University of Pennsylvania, Philadelphia, PA, USA. [5]The Jackson Laboratory, Bar Harbor, ME, USA. [6]European Molecular Biology Laboratory, European Bioinformatics Institute, Wellcome Genome Campus, Hinxton, Cambridge, UK. ✉e-mail: melania.bruno@nih.gov; todd.macfarlan@nih.gov

coevolution[10]. ERVs are the genetic remnants of retroviral integrations in the germline that when passed on through generations can lead to the rewiring of gene regulatory networks in the host genome across evolutionary time[11,12]. Whereas new ERV subfamilies can establish themselves either from new viral infections or by diversification from other ERVs, it is not known how new KZFP genes coevolve with new waves of ERV infiltration of the host genome. Genomic analyses have shown that KZFP genes evolve by gene duplications that give rise to highly repetitive KZFP gene clusters[13,14], however, the precise evolutionary dynamics of how these clusters respond to ERV colonization to promote the emergence of new KZFP genes are still poorly understood.

The mouse lineage has been recently colonized by several murine specific ERVs[15]. Concurrently, mice and other Eumuroida species have undergone dramatic expansions of their KZFP gene repertoire, with hundreds of species-specific KZFP units. In contrast, other clades, such as Hominoidea, exhibit much less extensive expansions (Fig. 1a, Supplementary Fig. 1a). There is growing evidence that different *Mus musculus* strains and even sub-strains possess KZFP gene variants that explain diverse regulation of specific ERV families and their nearby genes[9,16–23]. Although it is known that KZFPs serve as an important regulatory layer for ERV repression, the KZFP gene clusters in mice remain relatively understudied genomic loci. This is largely due to gaps

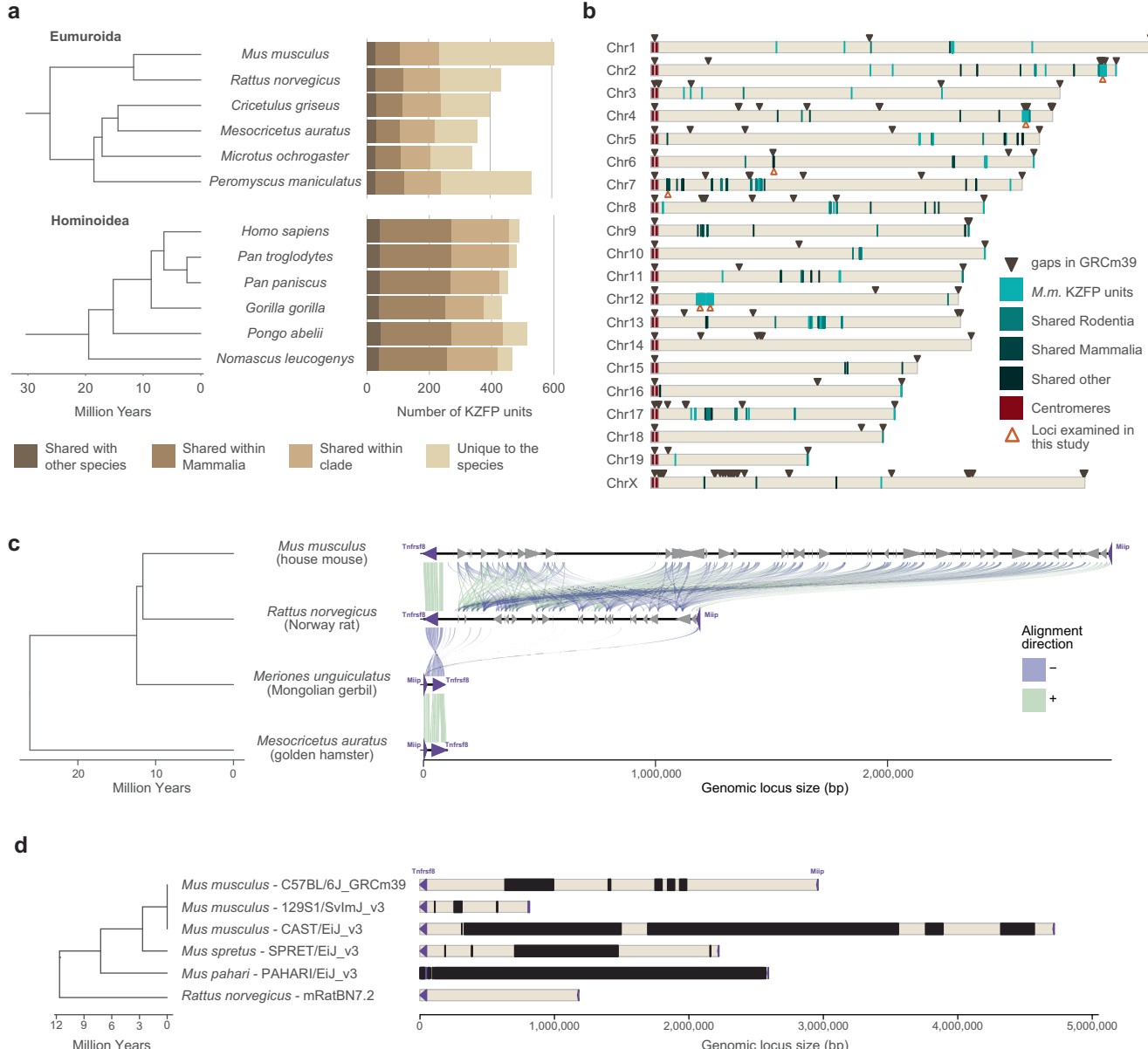

**Fig. 1 | Mouse displays several evolutionary young KZFP gene clusters with partially unknown sequence. a** Number of KZFP units (both coding genes and pseudogenes) in Eumuroida (top) versus Hominoidea (bottom) species, colored by conservation. From lighter to darker color: KZFP units unique to the individual species (compared to all species analyzed in Imbeault et al. 2017[1]); KZFP units shared with at least one other rodent (for Eumuroida) or primate (for Hominoidea) species; KZFP units shared with at least one other mammalian species besides rodents or primates; KZFP units shared with at least one other species besides mammals. The analysis extended to all species analyzed in Imbeault et al. 2017[1] and belonging to rodent and primate clades is shown in Supplementary Fig. 1a.

**b** Genomic distribution (GRCm39 assembly) of *Mus musculus* KZFP units, colored by conservation. Assembly gaps are indicated by dark arrow heads above the locus. The position of the KZFP gene clusters examined in this study is highlighted with orange arrow heads below the loci. **c** Comparison of the mouse Chr4 KZFP gene cluster locus between house mouse (GRCm39), Norway rat (GRCr8), Mongolian gerbil (Bangor_MerUng_6.1) and golden hamster (BCM_Maur_2.0). **d** Annotation of sequence gaps (black rectangles) in the Chr4 KZFP gene cluster locus in different mouse strains and species, and rat. The genes *Tnfrsf8* and *Miip* (up- and down-stream of the cluster, respectively) are shown as purple arrow heads.

in the available genome assemblies, which hinder comprehensive analysis of these loci and obscure the full extent of KZFP gene diversity and evolution across the mouse lineage.

In this study, we leverage the power of long-read sequencing technologies to investigate the content of young mouse KZFP gene clusters and uncover dynamics of their rapid evolution and divergence. We show that the integration of ERVs new to the mouse lineage within evolutionary young KZFP gene clusters likely promoted recombination events leading to the emergence of new KZFPs that repress them. Simultaneously, ERV copies also expanded by duplication in addition to retrotransposition, revealing a previously unrecognized mechanism driving ERV enrichment at these loci.

## Results

### De novo *Mus musculus* assemblies reveal a much larger KZFP gene cluster at the end of Chromosome 4

KZFP genes are organized in genomic clusters on several chromosomes in the *Mus musculus* genome (Fig. 1b). While some KZFP gene clusters primarily comprise old genes shared across species, others, such as the double-cluster on Chromosome 12 (Chr12), harbor genes entirely unique to mice. A few clusters, like those at the end of Chr2 and Chr4, contain at least one KZFP gene shared with other rodent species while mostly encoding KZFP genes unique to mice. The KZFP cluster at the distal end of Chr4 stands out for several reasons. This KZFP gene cluster appears to be specific to the Murinae clade, likely originating in the last common ancestor of rats and mice (Fig. 1c). Despite the conserved syntenic block defined by *Tnfrsf8* and *Miip* genes flanking this locus, the cluster is absent in other closely related muroids, such as gerbils (Fig. 1c, Supplementary Fig. 1b). Comparative analysis between mouse and rat reveals that this region expanded significantly in the mouse lineage, acquiring multiple new KZFP genes. Finally, this cluster has been repeatedly implicated in studies mapping modifier loci for variably regulated ERVs across mouse strains[9,16,23], making it an excellent model to explore the evolution and diversification of KZFP gene clusters. Extensive analyses of this locus have been hindered by persistent sequence gaps in this region even in the current GRCm39 reference assembly, the size and content of which have remained largely undefined (Fig. 1d, Supplementary Fig. 1c).

Thus, we generated de novo genome assemblies to fill the gaps in this locus and other young KZFP gene clusters for two widely used laboratory mouse strains C57BL/6 J (BL6J) and 129S1/SvImJ (129S1), by combining the high sequencing accuracy of PacBio HiFi sequencing with the ultralong reads of ONT sequencing (Fig. 2a). While our primary goal was to resolve gaps in KZFP gene cluster loci, the resulting assemblies achieved high overall quality as assessed by the Benchmarking Universal Single-Copy Orthologs (BUSCO) analysis[24], and with several contigs spanning entire chromosomes (Fig. 2b, c, Supplementary Fig. 2a, b). To refine the assemblies, we retained and strand-corrected contigs aligned to known locations in the GRCm39 reference, removing unplaced and redundant nested contigs (Supplementary Fig. 2a, b).

Focusing on the KZFP gene cluster on Chr4, we uncovered over 2.5 Mb of previously unassembled sequence in BL6J, resolving five gaps in the GRCm39 reference. This expanded the total cluster size from 2.8 Mb to 5.4 Mb and revealed new regions of high sequence similarity within this locus (Fig. 2d). De novo transcriptome assembly, combining published RNA-seq data[25] with newly generated PacBio IsoSeq data, allowed us to curate and complete the annotation of this locus. We identified 38 previously unannotated KZFP genes and resolved the full sequence of three genes (*Zfp986*, *Zfp993*, and *Gm21411*) whose zinc finger arrays were incomplete due to sequence gaps (Fig. 2d, Supplementary Data 1).

The identification of novel sequence nearly doubling the size of this locus underscores a key limitation of interpreting genomic datasets using the GRCm39 reference assembly, where short-read mapping of ChIP-seq and RNA-seq experiments can result in artificial read pileups when reads derived from gap sequences are misaligned to the highly similar sequences available in the reference assembly (Supplementary Fig. 2c, d).

Interestingly, the Chr4 KZFP gene cluster is even larger in the 129S1 strain than in BL6J, spanning 6.9 Mb and containing 83 KZFP genes (Fig. 2e, Supplementary Data 1). Accuracy of the de novo assemblies at the Chr4 KZFP gene cluster (as well as at another double cluster on Chr12) was confirmed by inspecting the alignment of strain specific long ONT reads (>100 kb) over these loci in the corresponding assembly (Supplementary Fig. 3).

Complete telomere-to-telomere (T2T) assemblies for C57BL/6 J and CAST/EiJ (CAST) mouse strains have recently become available[26], allowing us to extend our comparative analysis to a third mouse strain. Applying a similar gene annotation and curation strategy to the CAST Chr4 KZFP gene cluster, we identified 86 KZFP genes in a 6.6 Mb region (Supplementary Data 1). Importantly, despite differences in sequencing and assembly strategies, our BL6J assembly is structurally identical to the T2T C57BL/6J assembly at the Chr4 KZFP gene cluster, as well as at other young clusters (Supplementary Fig. 4). This suggests that the large differences in KZFP gene cluster size observed between the three mouse strains are due to strain specific, rather than individual, locus divergence.

### The Chr4 KZFP gene cluster has been independently expanding in different mouse strains and species

Sequence comparison of the BL6J, 129S1, and CAST Chr4 KZFP gene cluster revealed that this locus is fundamentally heterogeneous between the three mouse strains, beyond the size difference (Fig. 3a, b). While the beginning and end of the cluster are conserved, most of the locus is rearranged, with some regions of sequence similarity scrambled across the cluster and variably duplicated in the three strains. While the overall sequence comparison hints to high divergence of this locus in the three mouse strains, detailed comparison of the curated KZFP gene annotation in this cluster further revealed a disparate KZFP gene repertoire (Fig. 3c, d). Fingerprint amino acids - corresponding to the amino acids at the positions −1, +2, +3, and +6 within each zinc finger according to helical nomenclature - are major determinants of the KZFP DNA binding specificity as they directly contact the target nucleotide sequence. Thus, we focused on the arrays of fingerprint amino acids of the coding KZFP genes and identified the repertoire of distinct fingerprint arrays in the Chr4 cluster of each mouse strain (Supplementary Data 2). Several fingerprint arrays were shared across multiple KZFPs and we identified 38, 47, and 45 distinct Chr4 KZFP fingerprint arrays in the BL6J, 129S1, and CAST strains, respectively. Only a small number of fingerprint arrays were found to have an exact match across different strains, and instead, the majority of fingerprint arrays are unique to the individual strains (Fig. 3d). Even fingerprint arrays shared between strains often have different representation, with varying numbers of KZFP copies in each strain (Supplementary Data 2). Our analysis indicates that the Chr4 KZFP gene cluster has undergone parallel evolution in the BL6J, 129S1, and CAST mouse strains, marked by independent duplication events within the locus. This conclusion is further supported by the distinct patterns of self-identical sequences observed in the locus across the three strains (Fig. 3e).

Since the *Mus musculus* Chr4 KZFP gene cluster locus is much larger than the corresponding locus in rat, we investigated whether this cluster expansion is unique to *Mus musculus* or if it also occurred in other mouse species. To explore this, we generated a de novo assembly for *Mus spretus* using a partially inbred strain (SPRET2) and also extended our analysis to a de novo assembly of *Mus pahari*[27]. These assemblies completely spanned the locus corresponding to Chr4 KZFP gene cluster, as well as other KZFP gene clusters analyzed in this study, allowing comparison of gapless sequences. Sequence comparisons among *Rattus norvegicus*, *Mus pahari*, *Mus spretus*, and

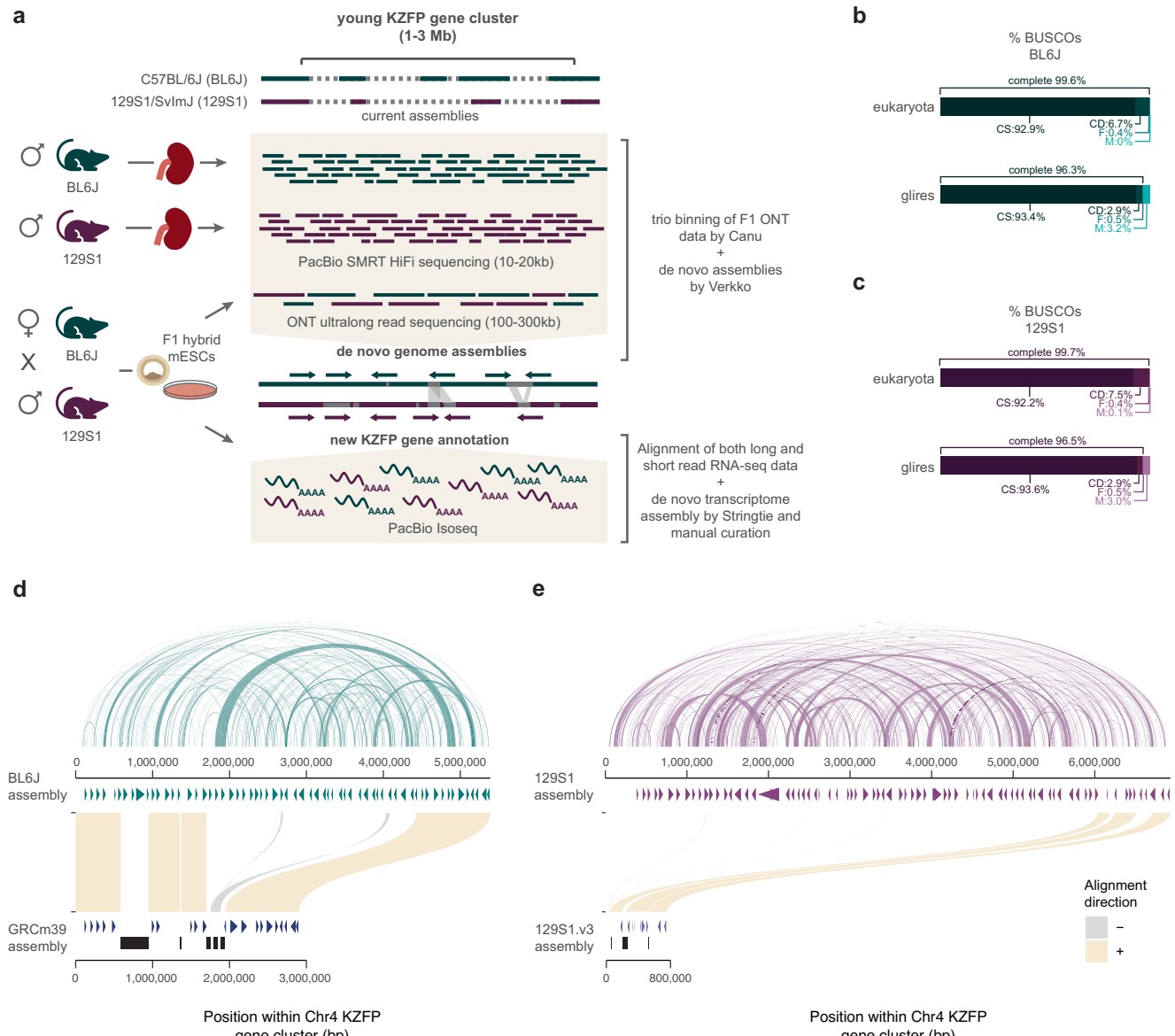

**Fig. 2 | De novo assemblies for the C57BL/6 J (BL6J) and 129S1/SvImJ (129S1) reveal that the Chr4 KZFP gene cluster is much larger than previously estimated. a** Scheme of the experimental setup and analysis to achieve the BL6J and 129S1 genome assemblies and curation of new KZFP gene annotation. **b, c** BUSCO statistics of BL6J (**b**) and 129S1 (**c**) de novo assemblies, after removing unplaced and redundant contigs. **d** Comparison of the Chr4 KZFP gene cluster (defined as 1 bp downstream of the *Tnfrsf8* gene and 1 bp upstream of the *Miip* gene) in de novo BL6J assembly with the reference GRCm39 assembly. Annotated genes are indicated as arrow heads. Gaps in the GRCm39 assembly are indicated in black. The arc plot on top of the de novo BL6J assembly indicates regions of self-identical sequence within the locus. **e** Comparison of the Chr4 KZFP gene cluster (defined as 1 bp downstream of the *Tnfrsf8* gene and 1 bp upstream of the *Miip* gene) in de novo 129S1 assembly with the available 129S1/SvImJ_v3 (GCA_921998555.2) assembly. Annotated genes are indicated as arrow heads; for 129S1.v3 assembly, genes from the Gencode M32 annotation lift-over are shown. Gaps in the 129S1.v3 assembly are indicated in black. The arc plot on top of the de novo 129S1 assembly indicates regions of self-identical sequence within the locus.

the three *Mus musculus* strains reveal that the independent expansion of the Chr4 KZFP gene cluster is a common feature of the mouse lineage compared to rat. However, the *Mus musculus* strains exhibit substantially larger cluster sizes than those observed in *Mus spretus* and *Mus pahari* (Fig. 3f).

**The Chr4 KZFP gene cluster rapidly expanded by large segmental duplications encompassing multiple KZFP genes**

To explore modes of rapid locus expansion, we traced the regions of segmental duplications within the Chr4 KZFP gene cluster. Due to the highly repetitive nature of this locus, many short sequence stretches exhibit high similarity, as revealed by various strategies for identifying self-identical sequences (Fig. 2d, e, Fig. 3e). However, detailed

annotation of the cluster's gene content in each strain showed that several genes shared identical or highly similar fingerprint arrays (Supplementary Data 2). This raised the question of whether these genes duplicated independently or as groups. To investigate this, we compared the sequences of the whole 3' exon (including both the portion encoding for the zinc finger array and the 3'UTR) of all KZFP genes within the BL6J cluster. This strategy enabled us to compare the underlying DNA sequence of the exon that contributes the most to the individual KZFP gene identity, while disregarding their actual coding potential. Limiting our analysis to the coding region would have biased the comparison, as truncated arrays caused by isolated point mutations would appear highly dissimilar, even though the sequences are nearly identical (Supplementary Fig. 5). This analysis allowed us to

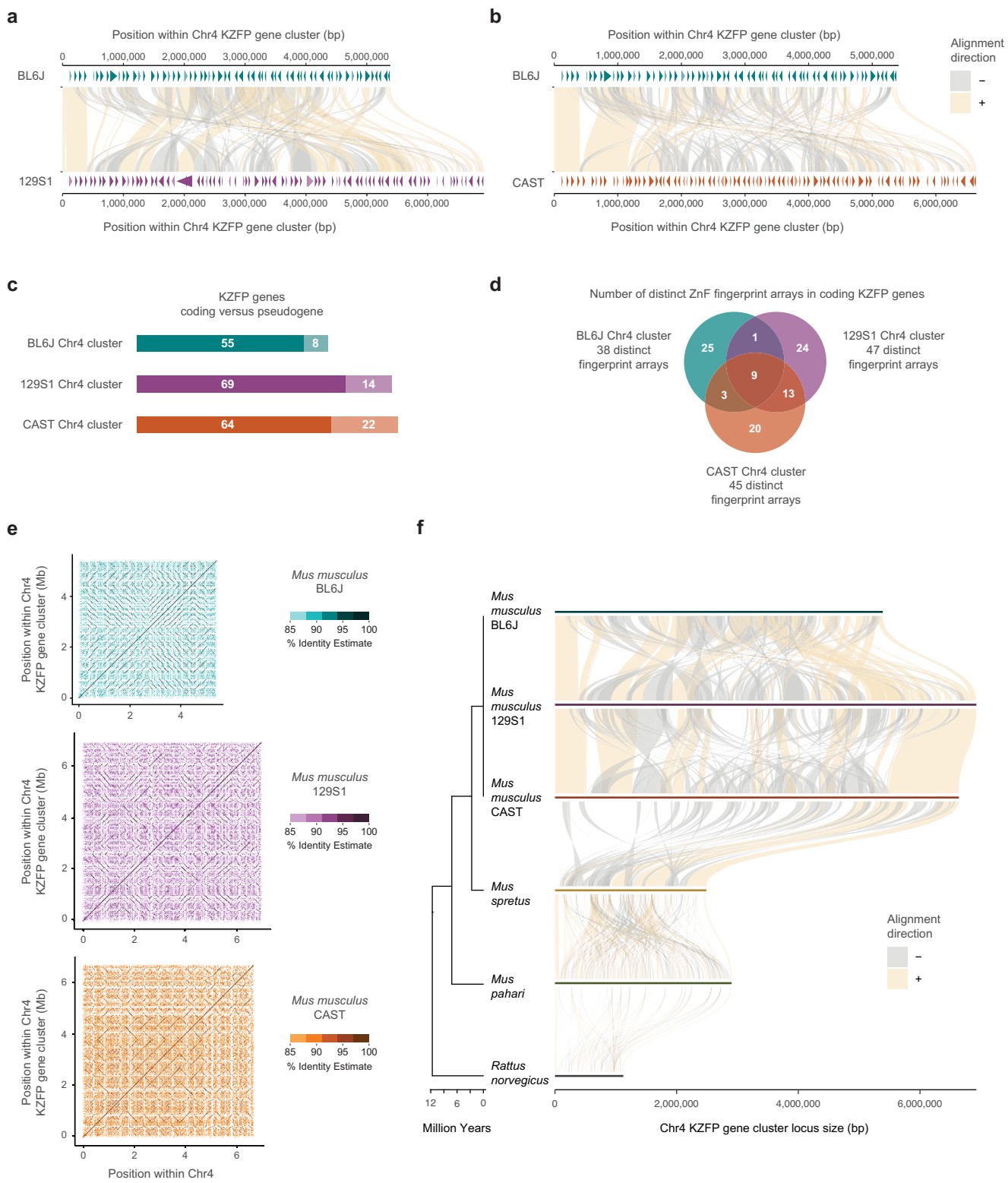

**Fig. 3 | The Chr4 KZFP gene cluster is highly heterogeneous between mouse strains and species. a** Comparison of the Chr4 KZFP gene cluster (defined as 1 bp downstream of the *Tnfrsf8* gene and 1 bp upstream of the *Miip* gene) in BL6J versus 129S1 mouse strains. Annotated KZFP genes, both coding (dark) and pseudogenes (light) after manual curation based on de novo transcriptome assembly, are displayed as arrow heads. **b** Comparison of the Chr4 KZFP gene cluster (defined as 1 bp downstream of the *Tnfrsf8* gene and 1 bp upstream of the *Miip* gene) in BL6J versus CAST (GCA_964188545) mouse strains. Annotated KZFP genes, both coding (dark) and pseudogenes (light) after manual curation based on de novo transcriptome assembly, are displayed as arrow heads. **c** Summary of KZFP gene content in the Chr4 cluster of the BL6J, 129S1 and CAST mouse strains. **d** Summary of the distinct fingerprint array content in coding KZFP genes in the Chr4 cluster of the BL6J, 129S1 and CAST mouse strains. The overlaps indicate 100% identity between fingerprint arrays. **e** Self-identity dotplots highlighting patterns of high sequence similarity within the Chr4 KZFP gene cluster in BL6J, 129S1 and CAST mouse strains. **f** Sequence comparison of the Chr4 KZFP gene cluster locus (defined as 1 bp downstream of the *Tnfrsf8* gene and 1 bp upstream of the *Miip* gene) between the three strains of *Mus musculus* with *Mus spretus*, *Mus pahari* and *Rattus norvegicus*.

identify similarity relationships between all the KZFP genes within the BL6J Chr4 cluster (Fig. 4a). By examining the gene positions within the cluster alongside their sequence similarity, we uncovered gene blocks - groups of genes with high similarity that were located in different regions of the cluster. These gene blocks ranged in size, containing between 3 and 7 genes (Fig. 4b). Interestingly, we also found evidence of partial duplications suggestive of multiple rounds of interstitial segmental duplications.

Collectively, this analysis revealed that the Chr4 KZFP gene cluster is a highly recombinogenic locus and that large segmental duplications have been responsible for the rapid expansion of the Mus musculus locus.

## Chromosomal position and meiotic recombination are not major drivers of KZFP gene cluster expansion and heterogeneity in mouse strains

We next sought to identify features that may have contributed to the recombination and segmental duplications driving the expansion and divergence of the Chr4 KZFP gene cluster in the mouse lineage. Several molecular mechanisms can be responsible for segmental duplications[28–30]. It has been observed that recombination and segmental duplication events in human are more frequent in genomic regions close to the subtelomeres[31,32]. Given the proximity of the Chr4 KZFP gene cluster to the telomeric region, we investigated whether its chromosomal position might have played a crucial role in its recombinogenic potential. To this end, we analyzed two additional young KZFP gene clusters: one on Chr2 located at a similar distance from the telomere as the Chr4 KZFP gene cluster, and a non-telomeric KZFP gene double-cluster on Chr12 located much closer to the centromere (Fig. 1b). Like the Chr4 cluster, the KZFP gene cluster at the end of Chr2 had several gaps in both the GRCm39 reference assembly and the 129S1.v3 assembly (Supplementary Fig. 6a–c). Surprisingly, we observed that after filling the gaps, the BL6J locus is slightly smaller than initially predicted in the GRCm39 reference. A comparison of this KZFP gene cluster across mouse strains revealed that the locus is nearly identical between BL6J and 129S1 Mus musculus strains but shows dramatic expansion in the CAST strain (Supplementary Fig. 6d, e).

In contrast, the KZFP gene double-cluster locus on Chr12 exhibited pronounced heterogeneity among the three examined Mus musculus strains (Supplementary Fig. 7). Notably, this double-cluster is specific to the mouse lineage and is entirely absent in rat and other species. Interestingly, this locus not only underwent independent expansion in mice, but each of the two clusters expanded independently as well, as demonstrated by the varying sizes of the clusters in different mouse strains and species (Supplementary Fig. 7d, e). Furthermore, the CAST strain displayed evidence of recombination between the two KZFP clusters within this locus, resulting in an inversion detectable by the change in orientation of the non-KZFP genes located between the clusters. These findings indicate that the chromosomal position of the KZFP gene clusters does not necessarily dictate the recombinogenic potential of these loci: the Chr2 KZFP gene cluster, which is located near the telomere, shows little heterogeneity between BL6J and 129S1 mouse strains, while the Chr12 double-cluster, positioned far from the telomere, exhibits high heterogeneity across all three mouse strains analyzed. Moreover, inbreeding of Mus musculus strains also does not appear to be the primary driver of KZFP gene cluster expansion. This is demonstrated by the striking expansion of the Chr12 double-cluster in Mus spretus.

Since meiotic recombination can also promote the emergence of structural variants[33,34], we next investigated whether the young KZFP gene clusters are enriched for meiotic hotspots, which could explain the frequent recombination events and locus heterogeneity observed between mouse strains. To address this, we looked at available datasets from BL6J and CAST mouse testes for PRDM9 binding, which determines the position of meiotic double strand breaks, and for DMC1 binding to single-stranded DNA (SSDS), which indicates loci undergoing DNA break repair during meiosis[35,36] (Supplementary Fig. 8). Young KZFP gene clusters in the BL6J and CAST strains do not appear to be significantly enriched for PRDM9 binding sites or DMC1-bound single-stranded DNA accumulation during spermatogenesis, which are rather excluded from these loci, compared to adjacent genomic regions. This is consistent with low meiotic recombination frequency observed at zinc finger gene and repeat loci also in human[37]. While low-frequency meiotic recombination events cannot be entirely excluded, meiotic recombination does not appear to be a major driver of the sequence divergence observed at these loci.

## Young and divergent KZFP gene clusters harbor a high load of self-identical sequences compared to older conserved clusters

Since the KZFP gene clusters on Chr4, Chr2, and Chr12 are all young clusters, we also examined two evolutionarily older KZFP gene clusters: one located on Chr6 and the other on Chr7 in Mus musculus. These older clusters harbor KZFP genes that are conserved across mammals and are located on the respective chromosomes at positions similar to the Chr12 double-cluster locus (Fig. 1b). Both older clusters (Chr6 and Chr7) exhibit remarkable sequence similarity not only between the mouse strains and species analyzed but also with rat (Supplementary Fig. 9a, c). Furthermore, these older clusters display a much lower load (Chr6) or even absence (Chr7) of self-identical sequences (Supplementary Fig. 9b, d), compared to the younger KZFP gene clusters. While large stretches of high sequence similarity in the young KZFP gene clusters are likely a consequence of recently duplicated gene blocks, young KZFP gene clusters also contain an abundance of small self-identical repetitive sequences compared to the older and more conserved KZFP gene clusters.

## Young mouse KZFP gene clusters expanded together with the infiltration of mouse ERVs at these loci

Repetitive sequences and TEs have been shown to contribute to structural variations, including large deletions, segmental duplications and chromosomal rearrangements[38–40] and previous studies have demonstrated that mouse KZFP gene clusters display TE enrichment, particularly enrichment of ERVs[14]. Thus, we hypothesized that the expansion and divergence of the young KZFP gene clusters might correlate with TE enrichment at these loci in the mouse lineage compared to rat. To address this, we compared the TE content across the genomes of rat, Mus pahari, Mus spretus, and the three Mus musculus strains. Genome-wide, we observed only minor differences in the overall abundance of different TE classes, with a subtle increase in LTR elements in the mouse lineage compared to rat (Supplementary Fig. 10a). However, when analyzing the TE content specifically within the Chr4 KZFP gene cluster, we found that the rat cluster is heavily enriched in LINE elements (45% of the cluster), whereas in mice, this locus acquired a higher LTR load (Supplementary Fig. 10c). A similar pattern was observed for the Chr2 KZFP gene cluster (Supplementary Fig. 10b). The LTR load was particularly striking at the Chr12 KZFP gene double-cluster locus in Mus spretus and in the three analyzed Mus musculus strains. This locus, which underwent the largest expansion in Mus spretus, is absent in rat and quite small in Mus pahari, where it displays a marked LINE-rich composition compared to Mus spretus and Mus musculus (Supplementary Fig. 10d). This trend is less prominent in the older KZFP gene clusters located on Chr6 and Chr7 (Supplementary Fig. 10e, f). Further analysis of the enrichment of individual LTR families at each examined KZFP gene cluster revealed that distinct mouse specific ERVs have colonized different young clusters, particularly in Mus musculus and Mus spretus compared to Mus pahari and rat (Supplementary Fig. 10g, Supplementary Data 3). In contrast, older KZFP gene clusters in mice displayed enrichment for many fewer LTR elements, which were mostly shared with rat. Finally, analysis of the

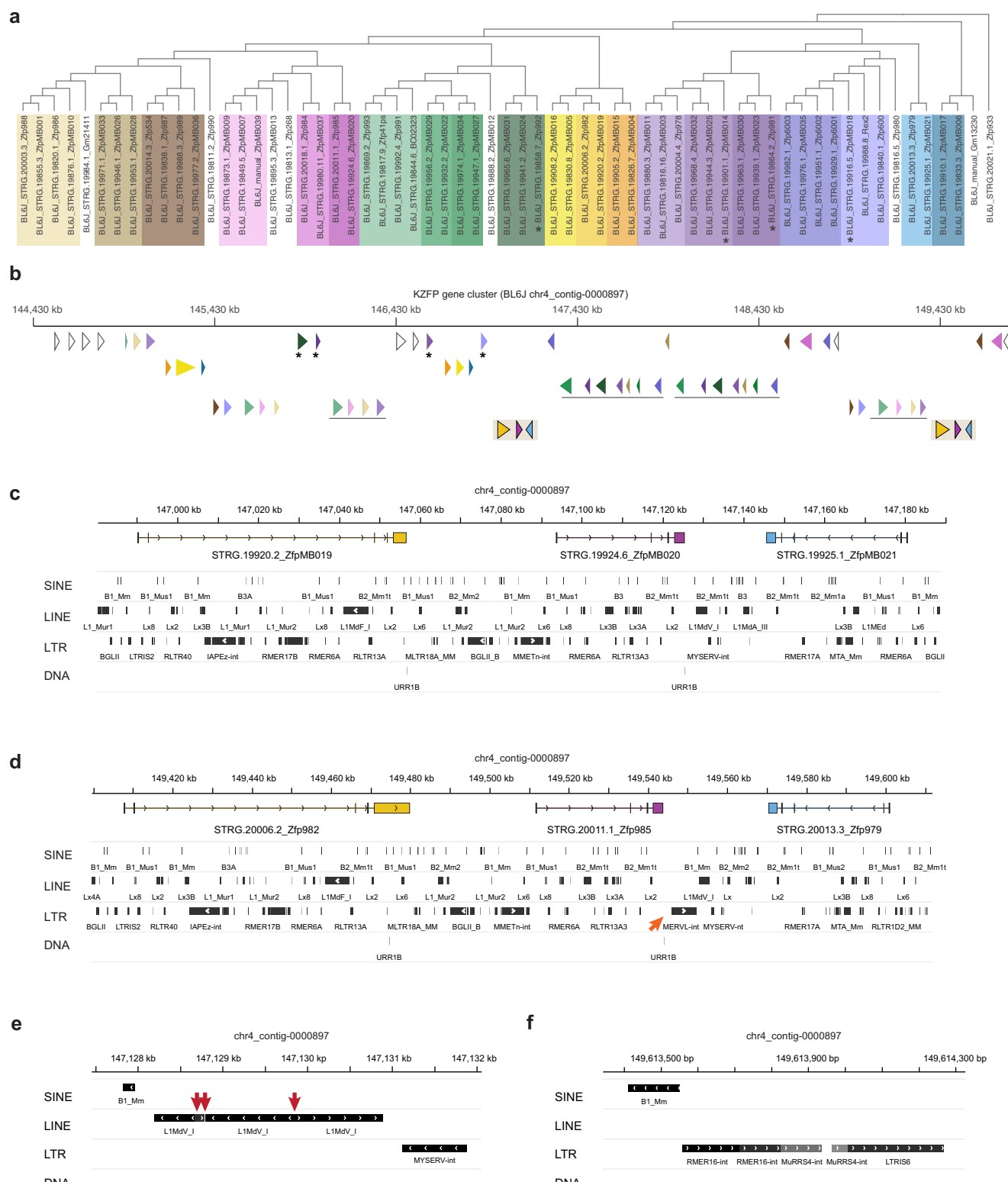

**Fig. 4 | The Chr4 KZFP gene cluster expanded by large segmental duplications spanning several genes and TEs. a** Tree based on multiple sequence alignment of the 3'exon of all the annotated KZFP genes (both coding and pseudogenes) in the BL6J Chr4 cluster. Same or similar colors highlight groups of KZFP genes with similar 3'exon sequence and clustered together in the tree. **b** Genomic position of KZFP genes in the BL6J Chr4 cluster. The genes are displayed as arrow heads and color coded according to the 3'exon sequence similarity analysis in (**a**). To enhance visibility, the annotation in distributed on multiple rows, with duplicated gene

blocks on the same row. Asterisks indicate isolated KZFP genes that share sequence similarity with KZFP genes in gene block duplication. The duplicated gene block shown in (**c**, **d**) is highlighted on the bottom row. **c**, **d** IGV snapshot of the duplicated gene block highlighted in (**b**). Genes are color coded to match the annotation in (**b**). TEs are shown at the bottom. The orange arrow indicates a new MERVL-int insertion in one of the duplicated gene blocks (**d**). **e**, **f** IGV snapshot of examples of chimeric LINE (**e**) and ERV (**f**) within the Chr4 KZFP gene cluster locus. The red arrows indicate breakpoints in the directionality of the chimeric LINE (**e**).

individual LINE families at the Chr4 KZFP gene cluster locus revealed that the LINE families with the highest representation at this cluster in mice are conserved at the corresponding rat locus (Supplementary Data 3), suggesting that this cluster might have originated as LINE-rich region and rapidly expanded in the mouse lineage together with the infiltration of new, lineage-specific ERVs. Close inspection of the Chr4 KZFP gene cluster revealed examples of chimeric LINE and ERVs (Fig. 4e, f, Supplementary Fig. 11). These chimeric elements have been described as the result of TE mediated non-allelic homologous recombination in which the initial retrotransposition event can also be the cause of the initial DNA breaks, and often associated with the establishment of structural variants[41–43], and we have found examples of these recombination scars at boundaries of recombined and duplicated portions of the BL6J Chr4 KZFP gene cluster (Supplementary Fig. 11). Altogether, our analysis suggests that the infiltration of new ERVs in the young KZFP gene clusters in the mouse lineage likely increased the frequency of non-allelic homologous recombination events, thus providing a possible mechanism whereby ERVs directly promote the expansion and divergence of young KZFP gene clusters.

### TE enrichment at young KZFP gene clusters is a consequence of the locus expansion by segmental duplication

Interestingly, while we observed increased content of mouse ERVs in several young KZFP gene clusters, we also found that the large duplicated gene blocks in the Chr4 KZFP gene cluster harbored copies of the TEs that had integrated prior to the segmental duplication events (Fig. 4c, d, Supplementary Fig. 11a). This suggests that the rapid gain of TE enrichment in KZFP gene clusters was driven by the segmental duplication events, rather than by retrotransposition events alone. Supporting this interpretation, we observed that despite the much smaller size of the *Mus spretus* Chr4 KZFP gene cluster locus relative to its *Mus musculus* counterpart, the overall enrichment for several ERV families remains similar between the two species (Supplementary Fig. 10g). This implies that the ERV load increased proportionally with locus expansion during duplication events in the mouse lineage. Further evidence for gain of TE enrichment independent of transposition comes from the enrichment of DNA transposons, prominent at the Chr12 KZFP double-cluster locus (Supplementary Fig. 10h, Supplementary Data 3). Because DNA transposons are mostly inactive fossil elements in mammals[44,45] - and even the few active ones replicate via a cut-and-paste mechanism rather than copy-and-paste - their increased enrichment suggests that pre-existing copies were duplicated during segmental duplication of the surrounding genomic regions.

To better understand these patterns, we focused on the BL6J Chr4 KZFP gene cluster, identifying specific ERVs that are highly represented relative to their genome-wide distribution. Of all the annotated MLTR18A_MM elements, 25.2% were found within the Chr4 KZFP gene cluster. Similarly, a large portion of all annotated ERVs of other distinct families (18.5% for LTRIS6, 16.1% for MMTV-int, 14.8% for RLTR13D3, 13.3% for LTRIS3, and 11.2% for RLTR1D2_MM elements) were found in the same cluster. These percentages are particularly striking, given that the Chr4 KZFP gene cluster accounts for only 0.2% of the total BL6J genome. Analysis of the sequence divergence of ERV subfamilies (measured as percentage of divergence to the consensus) for which more than 2% of total annotations occur at the Chr4 KZFP gene cluster revealed that while genome-wide there is a continuum of divergence (as expected for TEs that independently accumulate mutations) the Chr4 cluster displayed several groups of ERV copies with nearly identical sequence divergence (Fig. 5a). This is distinct from the overall bimodal distribution of sequence divergence observed for some ERV families[46], and likely reflects the segmental duplication of ERVs as opposed to independent insertions. Similar analyses in the 129S1 and CAST strains, as well as in *Mus spretus*, revealed strain-specific differences in ERV representation and distribution of sequence divergence

(Supplementary Fig. 11a, Supplementary Fig. 12), underscoring the dynamic nature of these loci.

Taken together, our findings suggest the following model (Fig. 5b): the integration of new ERVs within KZFP gene clusters may have increased the recombinogenic potential of these loci by promoting non-allelic homologous recombination events, driven by regions of microhomology shared between different ERVs. Repair processes leading to segmental duplications resulted in cluster expansion, possibly occasionally counterbalanced by repair events that caused cluster contraction. These two forces, operating independently and in parallel across different mouse strains, alongside species and strain specific ERV integrations, have likely driven the divergence of young KZFP gene clusters in mice. As a result, the loci now appear highly divergent, with the emergence of distinct young KZFP gene repertoires. This model would also explain how the content of ERVs and KZFP genes can increase in concert, with the KZFP gene clusters gaining more recombinogenic potential as they expand, due to the concomitant increase in repeat load. This raises an additional question: could the emergence of new KZFP genes that bind to and silence these newly integrated ERVs act as a brake on this self-reinforcing recombinogenic system?

### Several KZFPs in the BL6J Chr4 cluster target ERVs moderately enriched within the Chr4 cluster itself

To address whether there is a relationship between the ERVs that promoted KZFP gene cluster recombination and the emergence of new KZFP genes that could bind and repress them, we characterized the DNA binding properties of all the KZFPs encoded in the BL6J Chr4 cluster – with at least one amino acid difference – combining new ChIP-seq data for 42 KZFPs (including KZFPs with new or updated annotation) with previously published ChIP-seq data for 8 KZFPs[9]. Thus, we generated a TE target map for all the KZFPs encoded in the BL6J Chr4 cluster (Fig. 6a, Supplementary Data 5). As expected from previous studies, we observed that several KZFPs specifically target distinct TEs, and we were able to identify target motifs by integrating canonical motif discovery from experimentally determined peak regions with target motif prediction accounting for the KZFP amino acid sequence[47] (Supplementary Data 6). This analysis allowed us to identify the KZFPs responsible for targeting and silencing RLTR4 elements (Fig. 6b), that had been previously shown to be over-expressed in mESCs lacking the Chr4 cluster[9] and for which the specific KZFP responsible for their repression in wild-type cells had remained unknown. Furthermore, we found that many KZFPs exhibiting specific TE targeting were unique to the BL6J strain compared to 129S1 and CAST strains. Among them, we found several KZFPs that target distinct subfamilies of IAP LTRs and IAPEz internal regions (Fig. 6c), thereby identifying the modifiers likely responsible for reported variable methylation of these IAP elements in different mouse strains[16]. Lastly, we found examples of KZFPs that did not display strong or specific binding to TEs, and for which we could not even identify a general target motif (Supplementary Data 6), suggesting that not all the KZFPs that have emerged thus far in the BL6J Chr4 cluster have a specific function.

Interestingly, we also observed that several ERVs targeted by KZFPs encoded in the Chr4 cluster are moderately enriched at this locus, although they do not exhibit strong enrichment (Supplementary Fig. 10g, Fig. 6a). This observation hints to the intriguing possibility that the emergence of KZFPs targeting these ERVs may have acted as a brake on their enrichment. We speculate that KZFPs may limit the further expansion of their target ERVs in two ways synergistically: as KZFPs can repress the ERVs they bind to, they can reduce their retrotransposition; at the same time, as the ERVs cannot increase their numbers by new integrations, they cannot further increase the recombinogenic potential of the KZFP gene cluster locus, further limiting their expansion by segmental duplication.

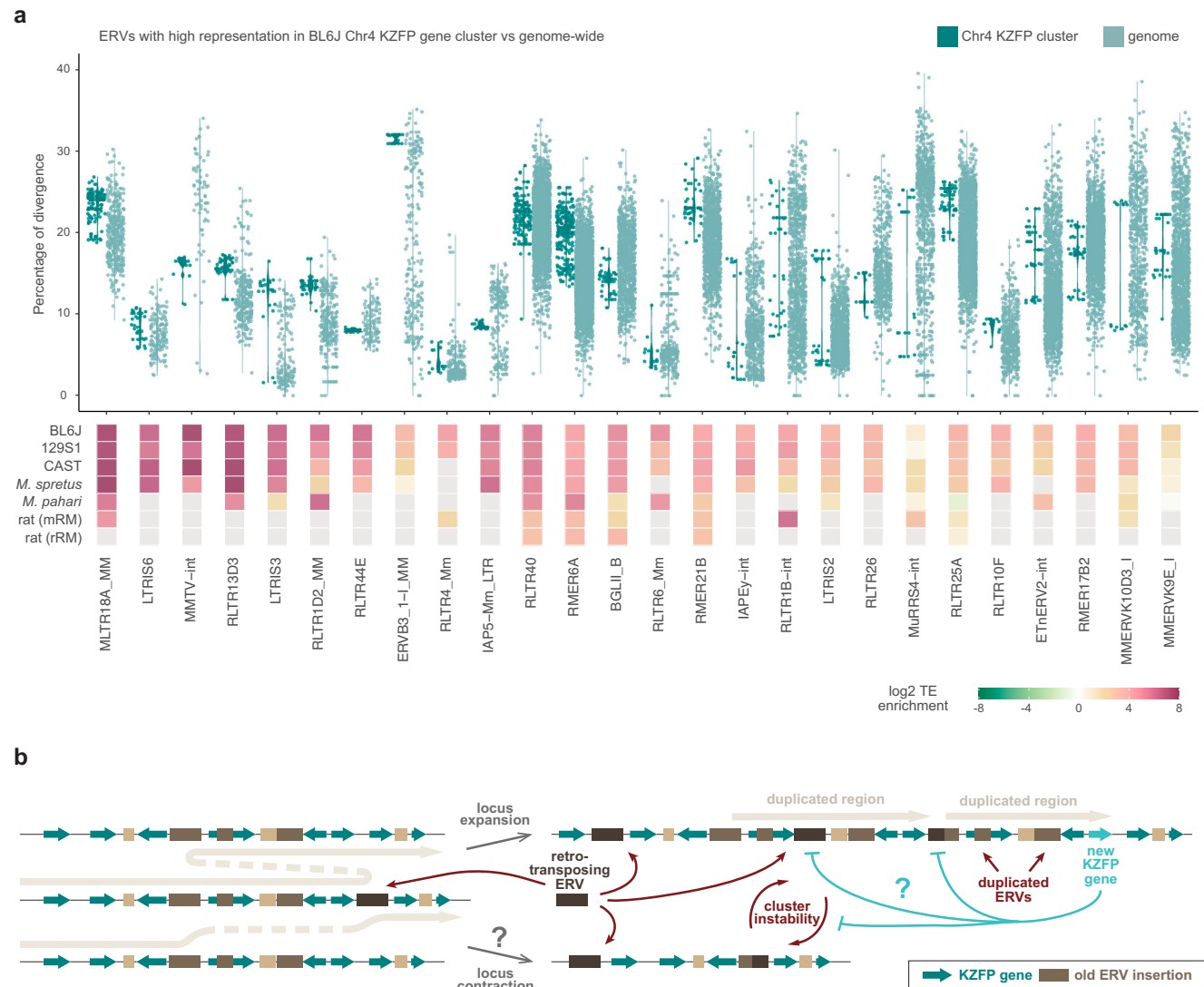

**Fig. 5 | Highly represented ERVs in the Chr4 KZFP gene cluster increased their number by duplication. a** Percentage of divergence from consensus of ERVs for which more than 2% of the total annotation occurrences are within the Chr4 KZFP gene cluster in BL6J (top), and their locus enrichment across the different *Mus musculus* strains, *Mus spretus*, *Mus pahari* and rat (bottom). ERVs are ordered based on the percentage of representation within the Chr4 KZFP gene cluster versus genome-wide in BL6J. ERV enrichment is shown as log2 of the ratio between the percentage of the KZFP gene cluster annotated as the ERV (bp) and the percentage of the whole genome annotated as the ERV (bp). Grey tiles indicate complete absence of the TE family in the corresponding locus. For rat, the enrichment using the mouse repeat annotation (mRM) and the rat repeat annotation (rRM) is shown. **b** Proposed model summarizing likely mechanisms of KZFP gene clusters expansion and diversification promoted by recombination between integrated ERVs.

## Discussion

KZFPs represent the largest family of DNA-binding factors in mammals, with a unique evolutionary flexibility in their overall numbers as well as in their DNA-binding specificity, which has enabled a remarkable diversification of the KZFP repertoire in different species. Rodents, and specifically mice, are an important model to understand how KZFP gene repertoires have been established over time, since KZFP genes have undergone dramatic expansion in different lineages, such as the Mus genus.

Although KZFPs have the potential to modulate the expression of ERV sequences, which are increasingly recognized for their role in both physiological and pathogenic processes[48–50], evolutionary young KZFP gene cluster loci in the mouse genome have remained relatively understudied due to the challenges associated with assembling these highly repetitive loci completely and reliably. In this study we demonstrate that the complete assembly and annotation of a single locus (Chr4 KZFP gene cluster) has led to the discovery of dozens of previously uncharacterized KZFP coding genes, which have the potential to contribute to complex gene regulation by repressing specific TEs. Furthermore, with the availability of full sequences from three different mouse strains, we were able to explore, at an unprecedented resolution, the significant heterogeneity of three young KZFP gene clusters in mice. Our findings reveal these loci as largely underexplored reservoirs of genetic diversity, tightly linked to ERV epigenetic heterogeneity. Although we could not observe large variation of the examined KZFP gene clusters between two BL6J *Mus musculus* individuals, we cannot exclude that pangenome analyses across larger populations and in different mouse strains and species might reveal more intra-species heterogeneity at evolutionary young KZFP gene clusters. Beyond the simple presence or absence of KZFPs with specific TE-targeting capabilities, we hypothesize that the varying numbers of copies of the same KZFP across different mouse strains could also influence transcriptional repression, contributing to gradual differences in gene regulation rather than binary on/off (gene

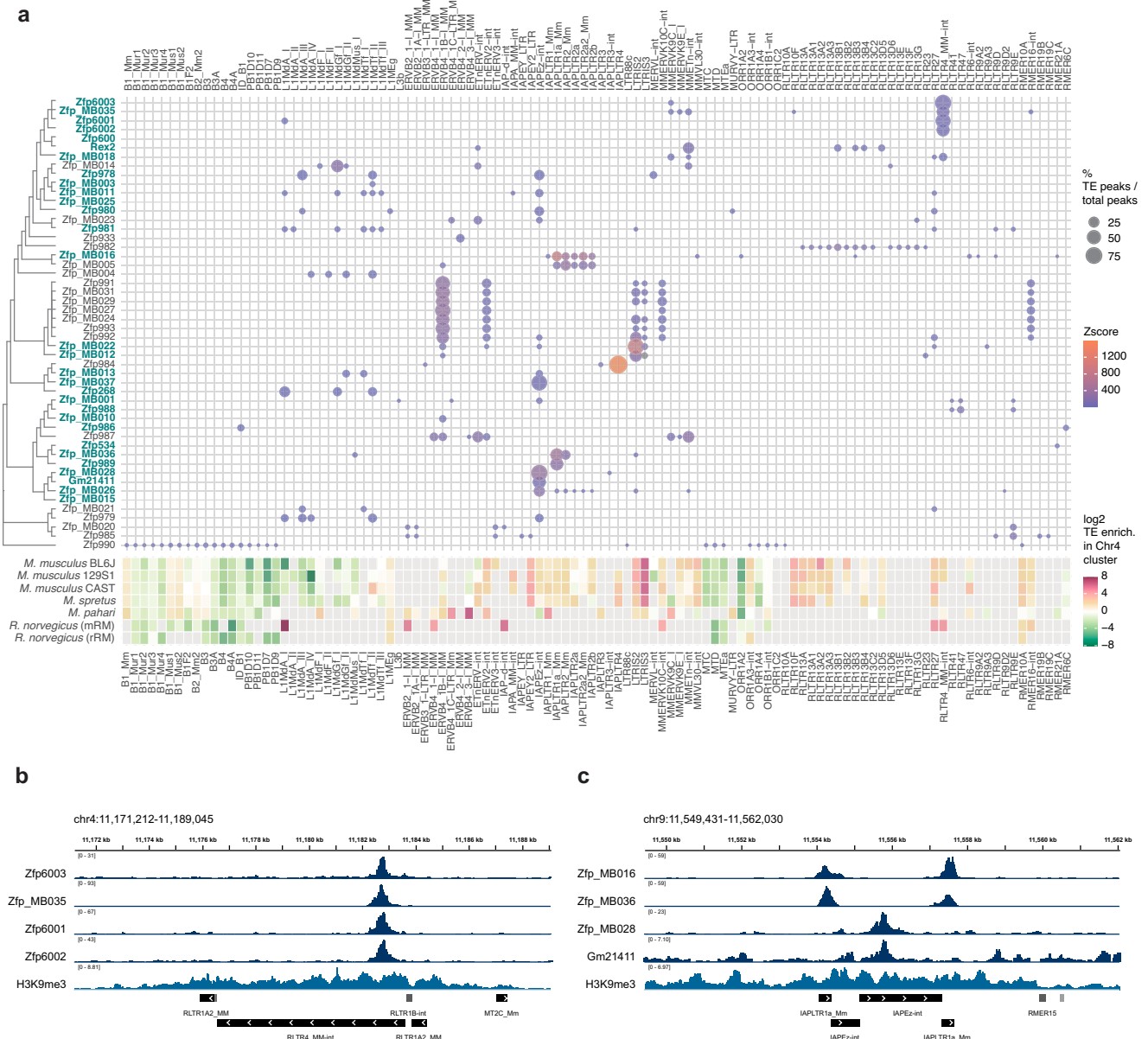

**Fig. 6 | Several KZFPs in the BL6J Chr4 cluster bind to ERVs that display only a mild enrichment within the Chr4 cluster itself. a** Bubble heatmap plot of TE binding of BL6J Chr4 cluster KZFPs (pValue < 0.001 from permutation test), determined by ChIP-seq experiments (top). Bubble size indicates the percentage of peaks overlapping with the distinct TEs over the total peaks for each KZFP; bubble color indicates the Z-score of the peak overlap over TEs, calculated by permutation test. KZFPs are sorted by similarity of the fingerprint array (tree on the left). KZFPs without an identical fingerprint array match in 129S1 or CAST strains are highlighted

in blue. The enrichment of each TE within the Chr4 KZFP gene cluster locus compared to genome-wide in different mice and rat is also shown (bottom). TE enrichment is shown as log2 of the ratio between the percentage of the KZFP gene cluster annotated as the TE (bp) and the percentage of the whole genome annotated as the TE (bp). Grey tiles indicate complete absence of the TE family in the corresponding locus. **b** IGV snapshot of multiple KZFPs targeting RLTR4_MM-int ERVs. **c** IGV snapshot of KZFPs targeting either the LTRs or the internal portion of IAPEz elements.

present or absent) states. Future studies that systematically annotate and functionally validate all KZFPs across different mouse strains will likely illuminate several observed differences in gene expression regulation. These differences are not easily explained by the mere presence or absence of individual genes[51], but might be due to the interplay of multiple KZFPs acting as epigenetic modifiers in more complex regulatory networks.

Our analysis also suggests that while new ERV integrations likely contributed to the expansion of KZFP gene clusters, their enrichment in these loci may be a consequence of the expansion of the KZFP gene clusters by segmental duplication, in a self-reinforcing loop that has likely promoted the independent expansion of evolutionary young

KZFP gene clusters in the mouse lineage and providing a compelling model for the correlation of LTR elements and KZFP gene numbers in the mouse genome compared to rat. The retroviral infiltration of KZFP genes might also not be a unique feature of the mouse lineage. ERV accumulation at KZFP gene clusters has also been observed in the Peromyscus lineage[52] and examples of lineage specific KZFP gene repertoire expansion are also present in primates[53]. A similar trend of ERV infiltration and KZFP gene cluster expansion is also evident in the human genome, where large primate-specific clusters are heavily infiltrated by primate-specific ERVs, in contrast to conserved KZFP gene cluster loci (Supplementary Fig. 13). One main question remains: do ERVs land at KZFP gene clusters only by chance, with no significant

negative consequences, allowing them to escape selection, or is there an active mechanism that facilitates the integration of new ERVs at these loci? While more work is also required to address how new KZFP gene clusters are seeded, we can start to speculate that certain TE integrations may have promoted recombination events between regular genomic regions and existing KZFP gene clusters, leading to the emergence of new clusters.

## Methods

### Analysis of KZFP unit conservation in rodents and primates
We re-analyzed the KZFP unit census (including genes, pseudogenes, and predicted related sequences) from Supplementary Table 2 of Imbeault et al. (2017)[1], using cluster IDs to represent distinct zinc finger arrays. To account for gene duplication events, we also considered the number of KZFP units with the same cluster ID in each analyzed species.

First, we identified and counted KZFP units unique to each species within the primate or rodent clades, considering all species for comparison. Second, we quantified KZFP units shared between at least two rodent species and absent in non-rodent species, and similarly, for primates. Third, we counted KZFP units shared with at least one other mammalian species but absent outside mammals. The remaining KZFP units represented those shared across species beyond mammals.

### Phylogenetic trees, ideograms and synteny plots
Phylogenetic trees were generated using TimeTree[54] and downloaded in Newick format. Trees were then visualized using the ggtree R package[55]. Ideogram plots were generated using the karyoploteR package[56], while synteny plots were generated using the SVbyEye tool[57].

### De novo assembly of C57BL/6 J and 129S1/SvImJ mouse strains and *Mus spretus*
PacBio HiFi sequencing was performed for both C57BL/6J (BL6J) and 129S1/SvImJ (129S1) mouse strains and *Mus spretus*. All mouse procedures had been reviewed and approved by the National Institute of Child Health and Human Development (NICHD) Animal Care and Use Committee (ACUC) at the National Institutes of Health (ASP#: 24-026). Mice were housed with a light cycle of 14 h on/10 h off, temperature maintained between 20 and 22 °C, humidity kept at 40–55%, and with no more than 5 mice per cage. Adult mice (between 2 and 6 months of age) were euthanized using $CO_2$ following the approved procedure.

Genomic high molecular weight (HMW) DNA was isolated from kidney of adult males of pure strain C57BL/6 J and 129S1/SvImJ mice purchased from JAX (Strain #000664 and #002448, respectively) and of an adult female of SPR2 Mus spretus (RIKEN RBRC00208), using the Monarch HMW DNA Extraction kit for Tissue (NEB T3060), following manufacturer instructions.

Sequencing libraries were prepared using the SMRTbell Express Template Prep Kit 2.0 and sequenced on a Sequel II using version 2.0 sequencing reagents. Circular consensus sequence (CCS)/HiFi reads were generated off-instrument from the initial subread data from each SMRTCell using the pb_ccs workflow (ccs version 6.3.0) within PacBio SMRTLink version 11.0.0.146107.

ONT ultralong read sequencing was performed for BL6J and 129S1 mouse strains, by deriving F1 hybrid mouse embryonic stem cells (XY) from E3.5 blastocysts from a cross between a female BL6J and a male 129S1 mouse. Cells were tested for karyotype stability by mitotic chromosome spreading and counting. Ultra-HMW genomic DNA was extracted with the Monarch HMW DNA Extraction Kit for Cells and Blood (NEB T3050) following manufacturer instructions. Libraries were prepared using the Nanopore Ultra-Long Sequencing Kit (SQK-ULK001) following manufacturer instructions and sequenced on an ONT FLO-PRO002 and FLO-MIN106 flowcells. Basecalling was done on instrument using Guppy v6.3.9, or Guppy v6.4.6 in high-accuracy mode.

To generate de novo assembly for BL6J and 129S1 mouse strains, first Canu was used for trio-binning of ONT reads[58], then Verkko was used to assemble each strain separately using the individual PacBio HiFi data as parental and the binned F1 ONT data[59].

De novo assembly of *Mus spretus* was generated using hifiasm[60] from PacBio HiFi data only, and the assembly graph of primary contigs output (.bp.p_ctg.gfa output file) was used to generate the assembly fasta file.

### De novo assembly QC and contig filtering
Assemblies were tested for completeness using BUSCO v5.4.7[61].

To facilitate navigation through the BL6J and 129S1 assemblies without scaffolding and the introduction of gaps, we retained only contigs that aligned to known chromosomes, using the GRCm39 assembly as a reference. De novo assemblies were aligned to the GRCm39 assembly using minimap2 with the -x asm5 option[62]. The resulting PAF files were filtered based on the following criteria: (i) only contigs aligning to known chromosomes were kept; (ii) if a contig aligned to multiple chromosomes, it was assigned to the chromosome with the longest alignment; (iii) for overlapping alignments, only the contig with the largest alignment was retained, removing smaller nested contigs. Additionally, strand corrections (flipping) were applied to contigs aligned to the minus strand. Due to fragmentation of ChrY in the BL6J assembly, with multiple contigs mapping to the same regions, only autosomes and ChrX contigs were retained for downstream analyses in both strains. The completeness of the filtered, strand-corrected assemblies was reassessed using BUSCO, confirming equivalent completeness compared to the unfiltered assemblies.

To assess assembly accuracy at the KZFP gene clusters, haplotype specific ONT reads (F1 ONT reads after trio binning using the parental PacBio HiFi data) were aligned to the corresponding assembly or, as a control, to the 'wrong' strain assembly using minimap2 (-ax map-ont option); the resulting sam alignments were converted to bam format and indexed using samtools view and index commands, respectively. Alignments of reads >100 kb were inspected in IGV to confirm tiling and homogeneous coverage of the read alignments over the KZFP gene cluster loci. Small indel threshold was set to 50 bp to improve visibility.

### SNP analysis between two BL6J assemblies
Sequence alignment was performed using the nucmer program, part of MUMmer tool (v 4.0.1)[63], using the BL6J de novo assembly in this study as reference and the T2T BL6J assembly (GCA_964188535) from[26] as query. The resulting delta.filter output was then used to identify SNPs with the show-snps command (with -Clr -T options), and SNPs within the examined KZFP gene clusters were quantified.

### Sequence alignments for synteny and self-identity plots
Sequence comparison of KZFP cluster loci was performed by pairwise sequence alignment using lastz[64] with (default options, --format=PAF) for the analysis across different mammal species shown in Fig. 1c and Supplementary Fig. 1b. For mouse versus rat comparison (Fig. 1c), only alignments larger than 3 kb were retained to improve visibility. For comparisons between different mouse strains, mouse species and rat minimap2 (-x asm20 -c --eqx --secondary=yes) was used. To compare KZFP gene cluster loci between de novo assemblies and available reference genomes (GRCm39 and 129S1.v3), the same minimap2 strategy was used, but with --secondary=no option. For self-identity arc plots, minimap2 was used with the following options: -x asm5 -c --eqx -D -P --dual=no. Self-identity arc plots above the alignment plots in Fig.2d, e were generated using SVbyEye, after filtering the paf file with filterPaf(., is.selfaln = TRUE). Self-identity dotplots were generated using the ModDotPlot tool[65] with the static -id 85 --color <custom colors> options. The -w parameter was set to 2000 for all the plots to maintain the same window size across assemblies and clusters,

with the only exception of the KZFP gene cluster on Chr7, for which -w was set to 500 to account for the much smaller locus size.

## Known gene and repeat annotation

Known genes from the Gencode M32 annotation release were annotated using the liftoff tool (with -copies option)[66]. This approach was also used to generate gene annotation and identify all KZFP gene cluster boundaries in the available assemblies from different mouse strains. Repeats were annotated using RepeatMasker v4.1.5[67] (-species "mus musculus") using NCBI/RMBLAST [2.14.1 + ] search engine and Dfam with RBRM v3.8.

## Manual curation of new KZFP gene annotation at the Chr4 cluster

Manual curation of de novo KZFP gene annotation at the Chr4 KZFP gene cluster in BL6J and 129S1 mice was achieved by combining PacBio Iso-Seq from our F1 hybrid mESCs (from BL6J x 129S1) and published short-read RNA-seq from pure strain mESCs for the two strains (SRR23065639, SRR23065640, and SRR23065641 for BL6J; SRR23065649, SRR23065650, and SRR23065651 for 129S1)[25].

For PacBio Iso-Seq, RNA was isolated from mESCs using the Zymo Research Quick-RNA Microprep kit following the manufacturer instructions. Iso-Seq libraries were created using the PacBio SMRTbell prep kit 3.0. Separate libraries were generated for mRNAs <3 kb or >3 kb. Sequencing was performed on a Sequel IIe sequencer (Pacific Biosciences) running instrument control software version 11.0.0.144466 and a movie collection time of 25 h per SMRTCell with 2 hr pre-extension. CCS/HiFi reads were generated from the initial subread data using the pb_ccs workflow (ccs v.6.3.0) within PacBio SMRTLink version 11.0.0.146107. CCS/HiFi reads with the proper orientation of 5' and 3' Iso-Seq primers on the sequence ends were identified using LIMA v.2.7.1. Iso-Seq primers were trimmed from the LIMA-FL reads. IsoSeq3 refine v.3.8.2 was used to evaluate and process the LIMA-FL reads by: (i) retaining reads with proper orientation of 5' and 3' IsoSeq primers and poly-A tail; (ii) orienting reads so that poly-A tail is on 3' end; (iii) trimming poly-A tail from read; (iv) removing possible chimeric reads (LIMA-FL reads with barcode sequences in the middle). Polished high quality isoform sequences (predicted accuracy >= 0.99) were obtained by using IsoSeq3 cluster on Refine-FLNC reads from both <3 kb and >3 kb mRNA libraries, to also reduce redundancy. These isoforms were used for downstream analysis and aligned to both BL6J and 129S1 assemblies using minimap2 (-ax splice:hq -uf options). Sam files were converted to bam using SAMtools v1.19[68].

BL6J and 129S1 short-read RNA-seq data was aligned to the respective genome assembly using STAR v.2.7.10b[69] with default parameters. De novo transcriptome assemblies for BL6J and 129S1 were generated using StringTie v.2.2.1 (--mix -u options)[70], combining the bam files from both short-read RNA-seq and PacBio Iso-seq data.

For the manual curation of Chr4cl KZFP genes in the CAST mouse strain the same analysis approach was used, combining published short-read RNA-seq data from pure strain mESCs (SRR23065636, SRR23065637, SRR23065638) with ONT long-read RNA-seq data instead of PacBio Iso-Seq. Briefly, RNA was isolated from F1 BL6J/CAST mESCs (XY) – described in ref. [26] – using the Qiagen AllPrepDNA/RNA Kit. ONT sequencing libraries were generated using the ONT cDNA-PCR Sequencing V14 - Barcoding (SQK-PCB114.24) and sequenced on a PromethION machine using the 10.4 flow cell chemistry. Fastq files were generated with the base caller Dorado with SUP (super accurate) setting. Reads were mapped to the CAST genome assembly (GCA_964188545) using minimap2 (-ax splice option).

For the manual curation of KZFP genes, all transcripts identified by StringTie and falling at the Chr4 cluster locus were manually inspected in IGV and compared with the bam alignment generated from the short-read RNA-seq for the specific mouse strain. We observed that all KZFP genes at the Chr4 cluster were highlighted by the presence of a 3'exon overlapping MMSAT4/MurSatRep1 repeats. The sequences from these transcripts were then imported into Snapgene and manually inspected for the presence of canonical splice sites around the identified exons and for open reading frames in the spliced transcripts. Whenever multiple transcript isoforms were detected for the same gene, we prioritized transcripts with higher abundance estimated by StringTie and inspected less abundant transcripts when the most abundant ones were missing critical exons that were obviously highlighted in the bam track form the short-read RNA-seq. Transcripts with the potential to encode for both a KRAB domain and at least one zinc finger in at least one of the StringTie identified isoforms were used to annotate coding KZFP genes. Transcripts missing the start codon or the exon encoding for part of the KRAB domain were considered pseudogenes or otherwise classified as 'other', if they could still retain a minimal coding potential. Finally, for the CAST annotation only, we manually extended the end of the 3'exon whenever the StringTie assembled transcripts revealed coding potential but were truncated in the coding portion of the 3'exon, resulting in otherwise truncated zinc finger arrays; the extended annotation always contained several more in frame zinc fingers and a stop codon, revealing complete open reading frames.

## Fingerprint analysis

During the manual curation of the KZFP gene annotation, all zinc finger arrays were also manually annotated. Fingerprint amino acid sequences were then extracted by position within the annotated zinc fingers (−1, +2, +3, and +6 positions, according to helical nomenclature). Furthermore, fingerprints in zinc fingers with mutations for at least one of the two cysteine or histidine were highlighted in red; fingerprints in zinc fingers with mutations for at least one of the other conserved structural amino acids (−12 F/Y, −3 F, +4 L) were highlighted in yellow.

To sort the KZFPs by similarity of the fingerprint arrays and facilitate comparison of their target sequence in ChIP-seq experiments (Fig. 6a), sequences of fingerprint arrays were compared by multiple sequence alignment using MAFFT v.7[71] with G-INS-1 progressive alignment method and a 0.5 value for align versus leave gappy regions. The fasta alignment was then used to generate a distance matrix based on maximum likelihood estimation using the phangorn v.2.12.1 R package (dist.ml function)[72], and the matrix was used to generate a tree estimation based on the Minimum Evolution Algorithm, using the fastme.bal function of the ape v.5.8 R package[73]. The pml (default parameters) and optim.pml (optNni=TRUE option) of the phangorn package were then used to optimize the tree topology using nearest-neighbor interchange.

## KZFP gene duplication analysis

To identify highly similar KZFP genes within the BL6J Chr4cl, the 3'exon DNA sequence of all the annotated KZFP genes (both coding and pseudogene/other) was used to generate a multiple sequence alignment using clustalO 1.2.4 with default settings[74]. The output tree was saved in Newick format and displayed using the ggtree R package[55]. KZFP genes with highly similar 3'exons were highlighted with similar colors and used to identify patterns of KZFP gene cluster duplications spanning multiple genes.

## TE content, enrichment, and divergence analysis

TE content analysis for each assembly was performed based on de novo repeat annotation by RepeatMasker[67] for all mouse strains and species, and using both available annotation of rat repeats as well as de novo annotation of mouse repeats in the Rattus norvegicus mRatBN7.2 assembly.

TE enrichment at KZFP gene clusters was calculated based on the number of bp annotated for each TE subfamily as follows: (bp of TE/bp of KZFP cluster locus) / (bp of TE/bp of whole genome).

P-value of the overlap of each TE subfamily with each KZFP cluster was calculated by permutation test ($n = 1000$) using the overlapPermTest function with default settings of the regioneR package[75]. All data from this analysis is available in Supplementary Data 3; Supplementary Fig. 8g, h display the log2 enrichment for TEs that display enrichment with P-value < 0.001 in at least one KZFP gene cluster in one or more Mus musculus strain or in Mus spretus. The same strategy was used for the human TE enrichment analysis and TEs enriched with P-value < 0.001 in at least one KZFP gene clusters examined were displayed in Supplementary Fig. 13.

To compare the sequence divergence of LTR elements in the Chr4 KZFP cluster versus genome-wide, the percentage of divergence value calculated by RepeatMasker was used. All data from this analysis is available in Supplementary Data 4; LTR elements displayed in Fig. 5a were selected as more than 2% of their total annotations (and more than 10 annotations) occurred within the BL6J Chr4 KZFP gene cluster.

## ChIP-seq experiments

ChIP-seq experiments were performed by over-expressing each KZFP coding construct into mouse F9 embryo carcinoma cells (ATCC, CRL-1720). Cells were grown in Dulbecco's Modified Eagle's Medium (DMEM 4.5 g/L D-Glucose, L-Glutamine, 110 mg/mL Sodium Pyruvate) (Gibco, 11995-065) supplemented with 10% Fetal Bovine Serum (EMD Millipore, ES-009-B), 1X GlutaMAX (Gibco, 35050-061), 1X Anti-Anti (Gibco, 15240-062), at 37 °C under 5% $CO_2$.

The CDS for each KZFP was codon optimized for expression in mammalian cells, synthesized and cloned with a C-terminal 3x-HA tag by Genscript into a Sleeping Beauty (SB) transposon-based vector harboring a puromycin resistance cassette[9].

F9 cells were transfected with individual KZFP encoding pSB plasmids together with the DNA transposase plasmid, using Lipofectamine 2000 reagent (Invitrogen, Cat#11668019) according to manufacturer instructions. 48 h after transfection, cells were grown for additional 48 h in medium supplemented with a final concentration of 1 μg/mL puromycin. Cell cultures were then expanded using regular culture medium and expression of KZFP constructs of the expected size was validated by Western Blot.

For ChIP-seq, cells expressing each KZFP construct or transfected with empty vector as negative control were harvested by trypsinization, resuspended in PBS, counted and fixed in 1% formaldehyde for 10 min at room temperature with gentle mixing. Fixation was quenched with 0.4 M glycine for 5 min at room temperature with gentle mixing. Cells were washed twice with cold PBS and cell pellet was frozen on dry ice and stored at −80 °C. Cells were lysed in cell lysis buffer (5 mM PIPES pH8.0, 85 mM KCl, 0.5% NP-40, 1X EDTA-free protease inhibitor cocktail (Roche, 5056489001)) for 10 min on ice and homogenized using type-B dounce homogenizer. Released nuclei were pelleted and lysed in nuclei lysis buffer (50 mM Tris-HCl pH8.0, 150 mM NaCl, 2 mM EDTA pH8.0, 1% NP-40, 0.5% Sodium Deoxycholate, 0.1% SDS, 1X EDTA-free protease inhibitor cocktail) to release chromatin. Chromatin was sonicated with a Bioruptor® 300 (Diogenode), nuclear debris were then pelleted at 20,800 g for 15 min at 4 °C and supernatant was used for chromatin immunoprecipitation. Sonicated chromatin from 30 million cells was used for each ChIP. Dynabeads™ Protein A magnetic beads (Invitrogen, 10002D) were incubated with anti-HA tag antibody (Abcam, ab9110) and washed in 0.5% BSA in PBS. ChIP was performed overnight on rotation at 4 °C. Beads were then washed once with low salt wash buffer (0.1% SDS, 1% Triton X-100, 2 mM EDTA pH8.0, 20 mM Tris-HCl pH8.0, 150 mM NaCl), twice with high salt wash buffer (0.1% SDS, 1% Triton X-100, 2 mM EDTA pH8.0, 20 mM Tris-HCl pH8.0, 500 mM NaCl), twice with LiCl wash buffer (250 mM LiCl, 1% NP-40, 1% Sodium Deoxycholate, 1 mM EDTA pH8.0, 10 mM Tris-HCl pH8.0) and twice with TE buffer (10 mM Tris-HCl pH8.0, 1 mM EDTA pH8.0). Beads were incubated overnight at 65 °C in elution buffer (10 mM Tris-HCl pH8.0, 0.3 M NaCl,

5 mM EDTA pH8.0, 0.5% SDS) with 0.1 μg/μL RNAseA (Thermo Scientific, EN0531). Eluates were transferred to fresh tubes and incubated for 2 h at 55 °C with 0.3 μg/μL proteinaseK (Roche, 3115852001). For the chromatin input sample, elution buffer was added to an aliquot of sonicated chromatin and the sample was treated similarly to ChIP samples. DNA was finally purified with the DNA Clean & Concentrator-5 kit (Zymo Research, D4004).

DNA libraries for NGS were obtained with the ThruPLEX® DNA-Seq Kit (Takara, R400676) with DNA Single Index Kit −12S Set A and B (Takara, R400695 and R400695), following manufacturer instructions. Input samples for each experimental batch were mixed in equal amounts at the library preparation step to generate batch specific input samples. Samples were sequenced as 100 bp paired end reads on an Illumina NovaSeq 6000 system.

H3K4me3 and H3K9me3 ChIP-seq experiments in pure BL6J cells were performed the same as above with the following modifications: HGTC8 cells[76] were used for the experiment; anti-H3K4me3 (Abcam ab8580) and anti-H3K9me3 (Abcam ab8898) antibodies were used for ChIP.

## ChIP-seq analysis

Read quality was assessed by fastQC v0.12.1. Reads were aligned to the Mus musculus GRCm39 reference genome assembly (GCF_000001635.27) for all the KZFP ChIP-seq experiments, using the Burrows-Wheeler Alignment (BWA) tool v0.7.17[77] (bwa aln and bwa sampe commands, default settings – only one best alignment was randomly retained in case of equally good multiple read alignments). Sam files were then converted into bam files with SAMtools v1.19[68], while removing eventually unmapped and duplicated reads, and retaining only primary alignments (samtools view -F 0×4,0×400,0 ×100,0×800 -b -h file.sam > file.bam). Bam files were sorted and indexed with SAMtools and converted to bigwig normalized to 1x genome coverage (RPGC normalization) for each sample with deepTools v3.5.4[78] (bamCoverage --bam file.bam -o file.bw -of bigwig --binSize 10 --effectiveGenomeSize 2521902382 --normalizeUsing RPGC --extendReads 200). ChIP bigwigs were further normalized by the input of the respective batch using deepTools (bigwigCompare -b1 ChIP.bw -b2 input.bw -o ChIP_input_ratio.bw -of bigwig --operation ratio –skipZeroOverZero --binSize 10).

The same analysis strategy was used also to analyze the H3K4me3 and H3K9me3 ChIP-seq data from HGTC8 cells and previously published PRDM9 ChIP-seq data (GSM1493404)[35]. To generate bigwig normalized to 1x genome coverage for reads aligned to the de novo BL6J or CAST assembly, we used --effectiveGenomeSize 2525297504 or 2630722264, respectively, calculated using the unique-kmers.py command of the tool khmer v2.1.1 (with -k 200)[79–81].

For the re-alignment of previously published antiDMC1 SSDS datasets (GSM1954835, GSM1954839 and GSM1954846)[36] to de novo BL6J and CAST assemblies, identification of ssDNA derived reads and genome alignment was completed using a published SSDS processing pipeline (https://github.com/kevbrick/SSDSnextflowPipeline)[82]. Reads from hybrid samples were aligned in replicate to each parental strain genome using the same parameters. Also in this case, only one best alignment was randomly retained in case of equally good multiple read alignments.

Peaks from the KZFP ChIP-seq experiments were called using MACS2 v2.2.7.1[83] (macs2 callpeak -t ChIP.bam -c input.bam -f BAMPE -g 2521902382). Peaks with even 1 bp overlap with peaks called in any of the negative control replicate samples were removed.

Peaks were further filtered to only retain the ones with qValue =<0.01 and fold enrichment over input of 10. For the samples that retained less than 20 peaks, fold enrichment over input of 5 was used as cutoff.

The same analysis pipeline was used to re-analyze data published in Wolf et al. 2020[9], with the following modifications for single read

samples: bwa samse command (instead of bwa sampe) was used to generate sam files and -f BAM option (instead of -f BAMPE) was used to call peaks by MACS2.

For target motif analysis, fasta sequences of 200 bp surrounding the summits of the retained peaks were extracted using getfasta command from BEDTools v2.31.1[84]. Motifs were then identified using meme-chip tool from MEME suite v5.5.5[85], as well as the RCADE tool[47]. Target motif prediction based on the KZFP amino acid sequences alone was also performed, using the Zinc Finger Recognition Code (ZiFRC) tool[86].

TE enrichment of ChIP-seq peaks for each KZFP was assessed by permutation test ($n = 1000$) using the overlapPermTest function with default settings of the regioneR package (Supplementary Data 5). To validate distinct TE targeting by KZFPs (top bound TEs in Supplementary Data 6), ChIP-seq reads were re-aligned to the consensus sequences of all mouse repeats in the dfam repeat library and the presence of bona fide peaks over TEs that displayed peak enrichment with $P$-value < 0.01 based on the permutation test was manually inspected in the Integrative Genomic Viewer (IGV)[87].

For the bubble heatmap plot in Fig. 6a, a more stringent cutoff of $P$-value < 0.001 based on the permutation test was used.

### Reporting summary

Further information on research design is available in the Nature Portfolio Reporting Summary linked to this article.

### Data availability

All raw data generated in this study has been deposited in the SRA database under the BioProject PRJNA1219187. The ChIP-seq data generated in this study, together with the corresponding processed data files (normalized bigwig coverage files and peak files), has been deposited as a GEO series in the NCBI GEO database under accession number GSE292055.

The Mus musculus C57BL/6 J de novo assembly after contig filtering and strand correction generated in this study has been deposited at DDBJ/ENA/GenBank under the accession JBPRBH000000000; the version described in this paper is version JBPRBH010000000 [https://www.ncbi.nlm.nih.gov/nuccore/JBPRBH000000000] The Mus musculus 129S1/SvImJ de novo assembly after contig filtering and strand correction generated in this study has been deposited at DDBJ/ENA/GenBank under the accession JBPRBI000000000; the version described in this paper is version JBPRBI010000000 [https://www.ncbi.nlm.nih.gov/nuccore/JBPRBI000000000]. The Mus spretus SPR2 de novo assembly generated in this study has been deposited at DDBJ/ENA/GenBank under the accession JBQVYN000000000; The version described in this paper is version JBQVYN010000000 [https://www.ncbi.nlm.nih.gov/nuccore/JBQVYN000000000].

The raw data used to generate the Mus pahari assembly is available in the SRA database under the BioProject PRJNA966193.

The RNA-seq data used in this study are available in the SRA database under the BioProject PRJNA923323. Additional KZFP ChIP-seq data used in this study are available as a GEO series in the NCBI GEO database under accession number GSE115287.

The T2T C57BL/6 J assembly used in this study is available in the NCBI Genome database under the GenBank accession number GCA_964188535.1 [https://www.ncbi.nlm.nih.gov/datasets/genome/GCA_964188535.1/]. The T2T CAST/EiJ assembly used in this study is available in the NCBI Genome database under the GenBank accession number GCA_964188545.1 [https://www.ncbi.nlm.nih.gov/datasets/genome/GCA_964188545.1/].

The PRDM9 ChIP-seq data used in this study is available in the SRA database under accession number SRX689499. The DMC1 SSDS data used in this study is available in the GEO database as GEO series GSE75419. Source data are provided with this paper.

### Code availability

All data analysis was performed using publicly available tools and code options used are described in detail in the methods section.

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

## Acknowledgements

We thank James Thomas, Alice Young, Shelise Brook and Morgan Park at the National Institutes of Health Intramural Sequencing Center (NISC) for generating the long-read sequencing data for the C57BL/6 J and 129S1/SvImJ mouse strains; and we thank Tianwei Li and James R. Iben at the Eunice Kennedy Shriver National Institutes of Child Health and Human Development (NICHD) Molecular Genomic Core for generating the PacBio HiFi sequencing data of Mus spretus and the Next Generation Sequencing data for ChIP-seq experiments. We are very grateful to Takashi Akera and Warif El Yakoubi for providing Mus spretus specimens, as well as to members of the Macfarlan lab, Zuzana Loubalova and Adam Phillippy for helpful discussions. This study utilized the computational resources of the NIH HPC Biowulf Cluster (http://hpc.nih.gov). This work was supported by the Intramural Program of the Eunice Kennedy Shriver National Institute of Child Health and Human Development DIR 1ZIAHD008933 (T.S.M.) and ZICHD008986 (R.K.D.) at the National Institutes of Health (NIH), the Wellcome Investigator Award 210757/Z/18/Z (A.C.F.S. and K.C.) and the NIH grant R35-GM130302 (B.E.B.).

## Author contributions

Conceptualization, M.B. and T.S.M.; Methodology, M.B., S.M.F., A.M. and K.C.; Investigation, M.B., S.M.F., A.M., K.C., D.E.W.C., R.K.D. and T.S.M.; Data sharing, G.A.L., C.W.G., B.L.D., B.E.B. and T.M.K.; Writing – Original Draft, M.B.; Writing – Review & Editing, M.B., D.E.W.C. and T.S.M. with input from all the authors; Funding Acquisition, R.K.D. and T.S.M.; Resources, T.S.M.; Supervision, M.B., R.K.D., A.C.F.S. and T.S.M.

## Funding

## Competing interests

The authors declare no competing interests.
