## [Transparent Peer Review file · Nature Communications]

Young KRAB-zinc finger gene clusters are highly dynamic incubators of ERV-driven genetic heterogeneity in mice

Corresponding Author: Dr Todd Macfarlan

Version 0:

Reviewer comments:

Reviewer #1

(Remarks to the Author)

I read the manuscript titled "Young KRAB-zinc finger gene clusters are highly dynamic incubators of ERV-driven genetic heterogeneity in mice" submitted to Nature Communications - I hereby provide comments that aim to strengthen its main conclusions.

The manuscript aims to use a set of new high-quality assemblies the authors produced for various Murinae genomes to probe hard-to-assemble regions for missing genes, specifically focusing on clusters of genes of the KRAB zinc finger family. By doing so they fill significant gaps in the GRCm39 mouse genome assembly and unveil that some of these gaps are enriched in protein-coding KZFPs. They then highlight the evolutionary history of a cluster located on the telomeric end of Chr4, illustrating its recent amplification and duplication of murine-specific KZFPs. Using correlative analysis coupled with new ChIP-seq data they generate, they attempt to understand if a relationship exists between the amplified KZFPs and their transposable element targets that could influence dynamics of the cluster growth.

The work is high quality and the manuscript well-written. The new assemblies I am certain will be most welcome and useful for the community. It is also very nice to see confirmation of KZFP amplification by segmental duplication, but as the authors themselves state, this is a new finding. The discussion in parts is interesting, and I enjoyed specifically the new hypothesis that KZFPs could cooperate to adjust expression of their targets instead of the usual 'all or nothing' view. I must however say that the authors overreach in terms of conclusions by strongly suggesting mechanisms that are weakly supported by the evidence they present. Please find below both minor and major comments that I hope the authors will consider to enhance their manuscript.

Methodology:

line 551 - "KZFP genes were identified by the presence of a 3' exon overlapping MMSAT4/MurSatRep1 repeats" - please be aware that MMSAT4 repeats do not always line up with zinc finger arrays, they are basically an artefact of repeatmasker found exclusively in the mouse genome and match some zinc finger arrays better than others, and it would be better to call zinc finger domains using HMMER than rely on this.

line 573 - "Sequences of fingerprint arrays to sort KZFPs by paralogous DNA binding specificity were compared by multiple sequence alignment using MAFFT v.7 [67] with G-INS-1 progressive alignment method and a 0.5 value for align versus leave gappy regions."

Using such a multiple alignment strategy for zinc fingerprints can lead to partial misalignment between zinc finger domains, where matches will be identified, for example, between the last position of a zinc finger domain with the first position of another zinc finger domain, or many other similar scrambles which on the whole multiple alignment can yield some nonsense and significantly skew or distort evolutionary relationships inferred from the resulting tree. Please confirm that no such issues were identified, and/or consider using an aligner that is using per-domain information in blocks matching zinc finger domains and not per amino acid.

For the PRDM9, can you confirm that multiple mapping reads have been excluded or not? This could (both ways!) confound the current conclusions if for example mappability is very low in the newly identified highly repetitive clusters found in multiple copies.

Major points:

1- The claims of mechanism by which young TEs impact cluster amplification are overreaching, and many parts of the text (including the abstract and introduction) overly suggest this by similar overreaching quotes. For example, line 83-84 states that the manuscript 'uncover mechanisms for their rapid evolution and divergence.' - I see no such evidence in the manuscript.

At the core of the matter is the enrichment calculations of specific TE families in KZFP clusters - while some parts of the manuscript are balanced and clearly state that this could be due to either new TE replication, or simple carry-over from segmental duplications, other parts of the manuscript heavily suggest or straight-up claim to understand or use a language implying mechanism - both options should be discussed there but refrain from strong conclusions. Similarly, claims of KZFPs slowing down the putative mechanism should be toned down unless they can be supported with experimental evidence, or make it abundantly clear that it is a potential theoretical model.

If authors want to strengthen their mechanistical claims, it would in my opinion require tracking and quantifying all TE copies to determine which are new insertions at new and different locus of the segmentally duplicated blocks, and which ones have been carried over. They provide one example of this in figure 4d so I am confident that this can be done. Statistics then calculated using only the new integrants to show that they disproportionately are found in the clusters would strongly enhance the manuscript.

In some ways this is related to figure 5, which shows tight clustering of cluster specific ERVs split by subfamily compared to the rest of the genome-wide distribution. The way I see this figure is that it is very likely the vast majority of these are carry over from segmental duplications, even in the case of bimodal distributions such as LTRIS2 (which to me only means there was initially two integrants present before segmental duplication, one older and one newer) - for example with an older integrant being present in the proto-cluster with some divergence to the subfamily as it is old, and another younger integrant finding its way to the same locus just before multiple rounds of segmental duplications.

It should also be noted somewhere in the manuscript that these large young KZFP clusters are so far only been found in a handful of genomes, mostly in rodents. What does this imply for the suggested mechanism?

"and new HIV1 integrations have been detected at KZFP genes in human patients"

Please note that HIV-1 does not integrate in the germline, and the above citation does not demonstrate an enrichment - KZFP genes are numerous and it is expected integrants of new TEs will land in them randomly. It might be best to remove this reference.

2- It would be reassuring to see more evidence of the cluster assemblies being 'right'. The T2T comparison with the new *mus musculus* assembly is excellent and convincing, but such a comparison is not available for the other 2 genomes. Some extended data showing, for example, use of publicly available hi-c data to confirm assembly quality of the cluster would significantly enhance trust we can put in figures that show that the clusters are indeed scrambled between species. BUSCO score is good to see but only confirms assemblies are high quality overall, but is not informative for the new difficult-to-assemble cluster regions which, by definition, have no orthologous genes. Other options would include showing very long read coverage in some parts of cluster 4 to show that reads span, or statistics of complete KZFPs in the locus being covered by a single read.

Minor points:

Figures:

figure 1b - missing red triangles in the visual legend (although they are explained in the text of the legend, but weird double standard to have since everything else is there)

figure 1c - overlap of blue and green lines makes it likely a lot of green is completely hidden by the blue, consider putting those on two separate tracks

figure 3a and b somewhat redundant with panel f - panel f also implies some evolutionary sequence which might or might not represent faithfully the common ancestor at each branch point, having a tree on the side might lower the risk of this potential misunderstanding.

figure 3c 55 protein-coding kzfps in c57bl6, and the sum in figure 3d counts 38 distinct arrays - however, figure 6 and the related text says 'all KZFPs in the cluster' (line 341) yet the figure 6 panel A contains only 50.

figure 3 panel d - definition of 'different' and 'distinct' might change these results quite a lot - line 147 states 'extremely disparate' - a panel in figure 3 showing the full clustering analysis of znf signatures displayed by groups would enhance the manuscript. Also using an exact match is a high threshold and doesn't imply functionally unique as it is unclear how many zinc finger domains participate in the binding for each of the identified kzfps - the variability might not be functionally meaningful (yet) - a clustered version of signatures in supplementary table 2 in some cartoon format would be nice to see.

Figure 3 panel e has very limited usefulness, is unreadable unless zoomed in at 400% and would be better sent to the supplementary figures.

figure 4 - it would be nice to have each track be exclusive for a segmental duplication unit - the second line seems to contain two different type of units. Also, do you get the same results here comparing zinc fingerprints with a threshold of similarity (>70, >80%)

Text:

It would be appropriate to say in the early parts of the manuscript that similarly large clusters of young KZFPs have not been found in most species so far. Mention of the human T2T genome (and potentially a reanalysis to search for new KZFP clusters?) would enrich the manuscript.

line 32: "Most KZFPs repress endogenous retroviruses (ERVs) and other retrotransposons"
As we don't have any data in other species, it would be more appropriate to say "Most KZFPs in human and mice have been found to repress endogenous retroviruses"

line 34: "How new KZFPs emerge in response to ERV invasions is currently unknown."
"in response" is leading, "in correlation" would be more appropriate, so far there has been no great example / proof of the ERV invasion preceding the appearance of a KZFP that can recognize it. Also see line 67, 70, among others

line 39: "Our data supports a model by which new ERV integrations within young KZFP gene clusters likely promoted recombination events leading to the emergence of new KZFPs that repress them."
The evidence presented here is weak and correlative - such a strong statement in the abstract is misleading.

line 57: "While a few evolutionary old and conserved KZFPs have been shown to play essential roles in core developmental processes like embryonic development [3], imprinting [4, 5] and meiotic hotspot determination [6, 7], the vast majority of KZFPs bind to and repress transposable elements (TEs) [1]."
This sentence can give to new readers the false impression that evolutionary old and conserved KZFPs do not bind and repress TEs, which is incorrect.

early introduction: there is currently no mentions of the arms race scenario hypothesis derived from the ZNF91/ZNF93 data/paper.

line 90 and throughout the manuscript - more precision of which assembly is discussed at any given point would enhance understanding of the manuscript.

line 98 - can you confirm that both assemblies of these genomes are high quality enough that the cluster wouldn't be missed?

line 132 - extended figure 3 to me looks identical assemblies, is 'largely comparable' the most appropriate here? To me it is a weaker qualifier than it could be

Line 133 - "This suggests that the large differences in KZFP gene cluster size observed between the three mouse strains are due to strain specific, rather than individual, locus divergence." - although I agree this is likely, this claim would be stronger if the authors could confirm that the DNA used for the new assemblies is from multiple individual mice.

line 179 - why use the full 3' end exon to calculate similarity here, as the zinc finger signature is usually more informative? Is there a reason to switch which metric is used in different parts of the manuscript? Can you provide a comparison between both and use either a single one, or some merge of both?

Line 301 - are those DNA transposons still active in these genomes? Here the language implies that the only way DNA transposons can increase in copy number is by segmental duplication, which is incorrect.

Line 320 'support the following model', 'suggest' would be more appropriate
Most suggested mechanisms in this section are weakly supported by the data presented, and so is the model in figure 5b which could use a few more question marks to make it clear that these links are suggested - this is the kind of figure that along the way gets interpreted literally and clouds the understanding of newcomers to the field in the long term.

line 354 - "suggesting that not all the KZFPs that have emerged thus far in the BL6J Chr4 cluster have a specific function." - here it is in reference to not finding a motif - do they have peaks outside of TEs? The language implies that the only function of KZFPs is to bind TE, which is not correct - some of these KZFPs very well could have other functions than bind and repress TEs

line 374 - "KZFP gene cluster loci" - it would be more appropriate to say 'evolutionary recent KZFP gene cluster loci'

Reviewer #2

(Remarks to the Author)

This is an interesting and well-articulated manuscript that reports important observations on KZFPs, the largest group of transcription factors encoded by higher vertebrates yet a remarkably understudied family of proteins. Previous work demonstrated that KZFPs target ERVs and other transposable elements, which explains their high degree of lineage- or even species-specificity. Here, Todd McFarlan and collaborators zoom on mice, which are known to be subjected to ongoing ERV infiltration. Consistent with the TE/KZFP coevolution model, the authors accordingly find that KZFP gene pools differ very significantly between mouse strains. Using frontline genomic technology, they dissect one particular KZFP gene cluster, unveiling interesting information not only on the heterogeneity of the KZFPs found in different mouse strains but also on its ERV content and the role that these genetic invaders likely play not only on the expansion of KZFP genes but also on their regulation, with KZFPs in return controlling these TEs and remodeling of the cluster resulting in alterations or ERVs copy numbers.

The work is very well conducted and provides important insights in a family of proteins, the biological importance of which stands to be increasingly recognized as researchers stop being deterred by the repetitiveness of transposable elements-embedded sequences and being misled by the unfounded belief that KZFPs are largely redundant. Still, and no less significantly, the present study gives an important warning about the additional degree of attention to be paid if one works in the mouse model.

I do not have significant criticisms about the approaches and the results or their discussion. However, I suggest that the authors attempt some parallels with primate KZFPs, stressing similarities and differences with rodents. For instance, they could focus on the fifty or so KZFPs that appear to be conserved between mouse and human, and comment on the organization of their respective genomic environments, and try to assess whether these have been subjected or immune to TE infiltration after they were first established.

Reviewer #3

(Remarks to the Author)

The authors explore the distribution of KZFP genes clusters in the entire genome of mouse and conduct comparative genomics with other rodents. They focus on a cluster on Chr4, specific to the Murinae clade, thus very recent in evolutionary terms. They generate de novo genome assemblies to fill the gaps in this locus and other young KZFP gene clusters for two widely used laboratory mouse strains C57BL/6J (BL6J) and 129S1/SvImJ (129S1). Through this process, they discover, annotate, and curate new genomic sequences. To these data, they add publicly available comparable data for CAST/EiJ (CAST). The researchers perform a comparative analysis of the KZFP genes cluster on Chr4 in the three strains and determine that it evolved under parallel evolution through duplications within the locus. For BL6J they conclude that large segmental duplications were responsible for the expansion of this locus. They determine no effect of chromosomal location on the extent of heterogeneity within the locus, contrasting the cluster on Chr4 with another two young clusters (one in a similar telomeric position and one in a centromeric position). Then they contrast these three young clusters with older clusters they found, noting an abundance of small self-identical repetitive sequences in younger compared to older clusters, and also greater divergence in younger compared to older clusters.

Next, they explore how transposable elements (TEs) influenced the evolution of KZFP and found chimeric LINE-ERV elements in these regions, suggesting that non-allelic homologous recombination involving these elements likely drove KZFP cluster expansion in mice. The authors suggest that when KZFP clusters grow through duplication events, they simultaneously accumulate more transposable elements, creating more potential recombination sites, essentially creating a positive feedback cycle where expansion facilitates further expansion. They speculate that the emergence of new KZFP genes that target these newly integrated ERVs eventually slows down this self-reinforcing system. Finally, to explore the proposed mechanism, they use ChIP-seq data to show a potential negative feedback mechanism of KZFP genes on repetitive elements.

This work represents a significant advance in our understanding of KZFP gene evolution and its relationship with transposable elements. The authors make two noteworthy contributions: first, they discover substantial novel sequence absent from reference genomes, showcasing the power of long-read sequencing technology; second, they provide compelling evidence that newly expanded KZFP genes directly repress ERV expression.

Their research delivers unprecedented resolution of young KZFP gene clusters in mice through complete assembly and annotation. By resolving previously unknown sequences—nearly doubling the Chr4 cluster size—and annotating dozens of previously uncharacterized KZFP genes, the study overcomes a critical technical limitation that has hindered understanding of these genomic regions.

The methodology is sound, data analysis rigorous, and conclusions well-supported by the evidence presented. This work builds upon earlier studies identifying KZFPs as ERV repressors while advancing understanding of KZFP diversity in variable ERV regulation and the role of repetitive sequences in structural variation. What distinguishes this paper is its comprehensive integration of long-read sequencing, detailed genomic annotation, and functional characterization. I have only a few minor comments outlined below.

- Minor Comments

+ The paper would benefit from including brief explanations of technical concepts when they're first introduced. For instance, when discussing "fingerprint arrays," a simple definition explaining that these are the specific amino acid positions within zinc fingers that determine DNA binding specificity (if I get it right!) would help readers from outside the field. Adding these concise explanations for specialized terminology would make the research more accessible to readers outside this specialized field.

+ It would be interesting to discuss whether the authors expect variability at the population level within the same species.

While long-read data with sufficient accuracy for this analysis are not yet available at population scale, making direct evaluation currently impossible, these large rearrangements appear to be relatively old events (even the more recent ones). In my view, this suggests we should expect further divergence within species, possibly contributing to regulatory differences between individuals of the same species.

Reviewer #4

(Remarks to the Author)

Version 1:

Reviewer comments:

Reviewer #1

(Remarks to the Author)

I am satisfied with the positive response to my comments and in my opinion the manuscript is ready for publication.

Reviewer #2

(Remarks to the Author)

Congratulations for a very nice work!

Reviewer #3

(Remarks to the Author)

I have read the revised manuscript and the rebuttal letter, and I am pleased to see all the improvements. I have no further comments.

Reviewer #4

(Remarks to the Author)

Response to the reviewers' comments

We thank all the reviewers for their time in reading our manuscript and in providing comments to improve it. We have integrated most of the reviewers' suggestions in our revised manuscript and provide here a detailed response.

Reviewer #1 (Remarks to the Author):

I read the manuscript titled "Young KRAB-zinc finger gene clusters are highly dynamic incubators of ERV-driven genetic heterogeneity in mice" submitted to Nature Communications - I hereby provide comments that aim to strengthen its main conclusions.

The manuscript aims to use a set of new high-quality assemblies the authors produced for various Murinae genomes to probe hard-to-assemble regions for missing genes, specifically focusing on clusters of genes of the KRAB zinc finger family. By doing so they fill significant gaps in the GRCm39 mouse genome assembly and unveil that some of these gaps are enriched in protein-coding KZFPs. They then highlight the evolutionary history of a cluster located on the telomeric end of Chr4, illustrating its recent amplification and duplication of murine-specific KZFPs. Using correlative analysis coupled with new ChIP-seq data they generate, they attempt to understand if a relationship exists between the amplified KZFPs and their transposable element targets that could influence dynamics of the cluster growth.

The work is high quality and the manuscript well-written. The new assemblies I am certain will be most welcome and useful for the community. It is also very nice to see confirmation of KZFP amplification by segmental duplication, but as the authors themselves state, this is a new finding. The discussion in parts is interesting, and I enjoyed specifically the new hypothesis that KZFPs could cooperate to adjust expression of their targets instead of the usual 'all or nothing' view. I must however say that the authors overreach in terms of conclusions by strongly suggesting mechanisms that are weakly supported by the evidence they present. Please find below both minor and major comments that I hope the authors will consider to enhance their manuscript.

Methodology:

line 551 - "KZFP genes were identified by the presence of a 3' exon overlapping MMSAT4/MurSatRep1 repeats" - please be aware that MMSAT4 repeats do not always line up with zinc finger arrays, they are basically an artefact of repeatmasker found exclusively in the mouse genome and match some zinc finger arrays better than others, and it would be better to call zinc finger domains using HMMER than rely on this.

We shared concerns about potential pitfalls of automated strategies for the annotation of KZFP genes. To avoid any mis-annotation of KZFP genes (both coding and pseudogenes), we preferred not to automate the annotation process and performed a manual curation of all the transcripts identified by de novo transcriptome assembly using StringTie, as stated in the methods section:

“For the manual curation of KZFP genes, the transcripts identified by StringTie were manually inspected in IGV and compared with the bam alignment generated from the short-read RNA-seq for the specific

mouse strain". Our IGV session comprised the StringTie transcripts, the lifted Gencode M32 annotation, the bam track from the short-read RNA-seq, the bigwig for H3K4me3 ChIP - for BL6J only - (to verify the presence of active promoters) and the RepeatMasker annotation tracks. We are aware that MMSAT4 and MurSatRep1 repeats do not always overlap with the 3' exon of KZFP genes, as some of these repeats are scattered around and not part of transcripts, and some older KZFP genes are not marked by the presence of MMSAT4/MurSatRep1. However, we observed that all the KZFP genes at the Chr4 cluster are highlighted by these repeats.

Since we did not filter the transcripts a priori based on the overlap for the 3' exon repeats, we have clarified this by rephrasing the sentences in our methods section as follows:

For the manual curation of KZFP genes, all transcripts identified by StringTie and falling at the Chr4 cluster locus were manually inspected in IGV and compared with the bam alignment generated from the short-read RNA-seq for the specific mouse strain. We observed that all KZFP genes at the Chr4 cluster were highlighted by the presence of a 3' exon overlapping MMSAT4/MurSatRep1 repeats.

line 573 - "Sequences of fingerprint arrays to sort KZFPs by paralogous DNA binding specificity were compared by multiple sequence alignment using MAFFT v.7 [67] with G-INS-1 progressive alignment method and a 0.5 value for align versus leave gappy regions."

Using such a multiple alignment strategy for zinc fingerprints can lead to partial misalignment between zinc finger domains, where matches will be identified, for example, between the last position of a zinc finger domain with the first position of another zinc finger domain, or many other similar scrambles which on the whole multiple alignment can yield some nonsense and significantly skew or distort evolutionary relationships inferred from the resulting tree. Please confirm that no such issues were identified, and/or consider using an aligner that is using per-domain information in blocks matching zinc finger domains and not per amino acid.

We agree that the strategy used is not appropriate for addressing evolutionary relationships between KZFPs (which we indeed addressed differently), but it provided a good "functional" comparison. Because of its caveats, the fingerprint comparison strategy described was indeed only used to sort the KZFPs on the Y axis of the plot in Fig. 6a. We have provided the whole fingerprint array sequences in the Supplementary Table 6, where KZFPs are ordered as in Fig. 6a to help the comparison of the arrays when navigating the ChIP-seq results.

We have modified the methods section by first providing the rationale for the multiple sequence alignment "To sort the KZFPs by similarity of the fingerprint arrays and facilitate comparison of their target sequence in ChIP-seq experiments (Fig. 6a)..." and then provide details of the multiple sequence alignment strategy.

For the PRDM9, can you confirm that multiple mapping reads have been excluded or not? This could (both ways!) confound the current conclusions if for example mappability is very low in the newly identified highly repetitive clusters found in multiple copies.

We were concerned about mappability as well. Since the KZFP gene clusters are so repetitive, they do indeed present low mappability if only uniquely mapping reads are retained and all multi-mapping reads are excluded.

For all ChIP-seq data we used the same read mapping strategy: for reads that could be mapped to more than one location, only one best alignment was (randomly) retained (BWA default option). Before drawing any conclusion about meiotic hotspots at the KZFP gene clusters we excluded a read mapping

bias with this strategy by double checking how the signal from histone ChIP experiments compares to the Prdm9 ChIP data, using the same analysis pipeline:

This data is similarly shown for the BL6J assembly in Supplementary Fig.2, where also the ChIP and input tracks are shown separately before ratio normalization.

We can clearly see signal corresponding to both H3K4me3 and H3K9me3 at the KZFP gene clusters, so we conclude that read mapping bias was not responsible for the lack of Prdm9 (or SSDS) signal at these loci. Importantly, low rates of meiotic recombination at ZNF genes and repeats have also been observed in human, and we have added this information in the main text:

This is consistent with low meiotic recombination frequency observed at zinc finger gene and repeat loci also in human [37] (Spence, J.P. and Y.S. Song, Inference and analysis of population-specific fine-scale recombination maps across 26 diverse human populations. Science Advances, 2019. 5(10): p. eaaw9206.) While we are not adding the histone tracks also in the current Supplementary Fig. 8 (previously Extended Data Fig. 6) to avoid overcrowding and because we already show the data for the Chr4 cluster in

Supplementary Fig.2, we have improved clarity in the methods sections and now explicitly stated in the ChIP-seq analysis section how multimapping reads were handled in our data analysis:

... “only one best alignment was randomly retained in case of equally good multiple read alignments.”
And “Also in this case, only one best alignment was randomly retained in case of equally good multiple read alignments.”

Major points:

1- The claims of mechanism by which young TEs impact cluster amplification are overreaching, and many parts of the text (including the abstract and introduction) overly suggest this by similar overreaching quotes. For example, line 83-84 states that the manuscript 'uncover mechanisms for their rapid evolution and divergence.' - I see no such evidence in the manuscript. At the core of the matter is the enrichment calculations of specific TE families in KZFP clusters - while some parts of the manuscript are balanced and clearly state that this could be due to either new TE replication, or simple carry-over from segmental duplications, other parts of the manuscript heavily suggest or straight-up claim to understand or use a language implying mechanism - both options should be discussed there but refrain from strong conclusions. Similarly, claims of KZFPs slowing down the putative mechanism should be toned down unless they can be supported with experimental evidence, or make it abundantly clear that it is a potential theoretical model.

We understand the reviewer's point and we have reworded some sentences in the manuscript where we agree with the criticism.

In lines 82-84, we explain the aim of the study “In this study, we have leveraged on the power of long-read sequencing technologies to investigate the content of young mouse KZFP gene clusters and uncover mechanisms for their rapid evolution and divergence”. We believe we are not misleading the readers, since we are stating the purpose of our analysis and we do describe in the paper the presence of very large segmental duplications (spanning hundreds of kilobases and several genes and TEs at a time) as a mechanism that has allowed the rapid expansion of the Chr4 KZFP gene cluster in mice. We think Figure 4c-d convincingly shows an example of the large segmental duplications, now further backed up by the analysis in new Supplementary Fig. 11. However, to avoid semantic disagreements, we have changed “mechanisms for” to “dynamics of” at the end of the introduction section.

Regarding the TE enrichment calculations, we have improved the heatmap plots to distinguish lack of enrichment from complete absence of the TEs at the examined loci. This is a very important point that might not have come across. Indeed there are two connected but distinct conclusions that we make: 1) Young KZFP gene clusters, which also expanded in the mouse lineage, were infiltrated by many families of ERVs that are unique to the mouse lineage itself. While de novo infiltration by retrotransposition certainly still happens, by tracking the presence and absence (independently of the enrichment) of individual ERV subfamilies in rat, *Mus pahari*, *Mus spretus* and the 3 *Mus musculus* strains, we conclude that a massive colonization of the young KZFP gene clusters happened concomitantly with the clusters' expansion. The Chr12 double cluster locus is the best example, which presents a massive infiltration of Mus ERVs in *Mus spretus* and *Mus musculus* (where the locus has dramatically expanded) compared to rat and *Mus pahari*, where the clusters are either absent or very small. We have enhanced clarity on this point by distinguishing lack of enrichment (according to the color scale) from the complete absence of the TE family (grey) in all our TE enrichment heatmaps. Importantly, older and conserved clusters of KZFPs were generally not colonized by young ERVs in any lineage. This supports the model that the dramatic expansion of the young KZFP gene clusters likely happened concomitantly or right after the infiltration of these ERV families at these loci.

2) Together with KZFP genes, TEs also got duplicated in the large segmental duplication chunks contributing to the gain of enrichment at the young clusters (compared to their genome-wide distribution). This point was supported by the plots in Fig. 5a and Supplementary Fig. 12, and now further supported by the new Supplementary Fig. 11. We think this is an important point because the presence of duplicated TE copies provides information about the timing of TE integration. One striking example is given by RLTR4 elements. These LTRs are found at the Chr4 cluster only in the BL6J and 129S1 strains, but not in CAST or *Mus spretus*, suggesting that they are very recent. However, we found multiple RLTR4 copies imbedded in segmentally duplicated chunks, sharing the same distance relationship with copies of other ERV subfamilies. This demonstrates that the infiltration, while recent, happened prior to those duplication events in the BL6J and 129S1 cluster, which then led to the enrichment of this ERV subfamily in the locus.

About the role of KZFPs in potentially dampening the ERV-mediated recombination, we have been more careful with our wording:

“This observation hints to the intriguing possibility that the emergence of KZFPs targeting these ERVs may have acted as a brake on their enrichment. We speculate that KZFPs may limit the further expansion of their target ERVs in two ways synergistically: as KZFPs can repress the ERVs they bind to, they can reduce their retrotransposition; at the same time, as the ERVs cannot increase their numbers by new integrations, they cannot further increase the recombinogenic potential of the KZFP gene cluster locus, further limiting their expansion by segmental duplication.”

We and many others have previously shown that KZFPs repress the activity of the ERVs they bind to, and we don't see a reason why this should not apply also to ERVs that are present within the KZFP gene clusters themselves and mitigate further spreading of those ERVs.

If authors want to strengthen their mechanistical claims, it would in my opinion require tracking and quantifying all TE copies to determine which are new insertions at new and different locus of the segmentally duplicated blocks, and which ones have been carried over. They provide one example of this in figure 4d so I am confident that this can be done. Statistics then calculated using only the new integrants to show that they disproportionately are found in the clusters would strongly enhance the manuscript.

In some ways this is related to figure 5, which shows tight clustering of cluster specific ERVs split by subfamily compared to the rest of the genome-wide distribution. The way I see this figure is that it is very likely the vast majority of these are carry over from segmental duplications, even in the case of bimodal distributions such as LTRIS2 (which to me only means there was initially two integrants present before segmental duplication, one older and one newer) - for example with an older integrant being present in the proto-cluster with some divergence to the subfamily as it is old, and another younger integrant finding its way to the same locus just before multiple rounds of segmental duplications.

The reviewer's interpretation of Fig. 5a exactly mirrors our second major point and we are happy it came across: the enrichment of those new ERV families (once they infiltrated the loci) is a consequence of the segmental duplications.

The proposed analysis to track and quantify all TE copies to determine which are new insertions and which are resulting from duplications would certainly provide further support to our conclusions, however this analysis is not trivial if not impossible, since the whole Chr4 cluster resulted from interstitial rounds of duplications that make the extensive analysis unfeasible: even gene blocks that are not the immediate duplication of each other are highly similar to portions of other blocks, while recent duplications consist of portions of older duplications. We are showing this in the new Supplementary Fig.11. This makes it impossible to prioritize sequence comparisons to understand the full history of duplication and recombination events that have led to the current locus architecture. We have performed several attempts at reconstructing the order of events leading to the current Chr4cl sequence organization, but due to the high repetitiveness of the locus (and perhaps some missing information, as we cannot exclude deletion events as well!) we could not conclusively reconstruct the sequence of recombination events. We

concluded that this might be achieved only by sequencing samples from the different mouse species and strains harvested at different timepoints during their diversification (over millions or hundreds of years) - which is not feasible – or by building a new analysis tool that takes into account the positional pattern of TE integrations, which could be the focus of a completely new study.

The example shown in Fig.4d represents the only MERVL-int insertion present in the whole cluster, so it was an obvious example of a de novo integration after duplication, since there are no additional copies as also shown in Supplementary Fig. 11.

While we cannot extensively quantify the number of original integrant and duplicated copies for all the ERV subfamilies in the Chr4 cluster, we have done our best to perform a manual analysis showing some ERV subfamilies for which we could compare the *Mus spretus* and the BL6J *Mus musculus* assemblies to attempt to track their duplication history (new Supplementary Figure 11a). Furthermore, the patterns of different ERV subfamilies in relation to each other further highlight that the multiple copies are the result of segmentally duplicated blocks.

It should also be noted somewhere in the manuscript that these large young KZFP clusters are so far only been found in a handful of genomes, mostly in rodents. What does this imply for the suggested mechanism?

We think that the integration of ERVs into KZFP gene clusters may be a more general mechanism contributing to KZFP gene evolution across mammals, and we include another example in humans (see response to reviewer 2). We have added a new Supplementary Fig. 13 to show the KZFP gene cluster genome distribution in humans. The largest KZFP cluster in humans is ~4.5Mb, which is shared with other primates. Interestingly, this cluster is also enriched with primate-specific ERVs, suggesting infiltration of the KZFP cluster in a Hominoidea ancestor concomitant with the expansion of lineage-specific KZFP clusters.

"and new HIV1 integrations have been detected at KZFP genes in human patients"

Please note that HIV-1 does not integrate in the germline, and the above citation does not demonstrate an enrichment - KZFP genes are numerous and it is expected integrants of new TEs will land in them randomly. It might be best to remove this reference.

We have removed this sentence.

2- It would be reassuring to see more evidence of the cluster assemblies being 'right'. The T2T comparison with the new *Mus musculus* assembly is excellent and convincing, but such a comparison is not available for the other 2 genomes. Some extended data showing, for example, use of publicly available hi-c data to confirm assembly quality of the cluster would significantly enhance trust we can put in figures that show that the clusters are indeed scrambled between species. BUSCO score is good to see but only confirms assemblies are high quality overall, but is not informative for the new difficult-to-assemble cluster regions which, by definition, have no orthologous genes. Other options would include showing very long read coverage in some parts of cluster 4 to show that reads span, or statistics of complete KZFPs in the locus being covered by a single read.

We agree that assembly accuracy of the KZFP gene clusters is paramount. As suggested, we have added a new figure (current Supplementary Fig.3) demonstrating that long ONT reads (>100kb) tile the entire KZFP gene cluster locus on Chr4 for both the BL6J and the 129S1 cluster. We also show similar IGV snapshots for the Chr12 double cluster, which we also found to be highly heterogeneous between the two strains. Importantly, the BL6J Chr12 locus did not present gaps in the GRCm39 reference assembly and

our de novo assembly nicely matches the reference locus (as shown in current Supplementary Fig.7b). This locus serves as a good positive control for observing read pileups on a correctly assembled locus. To supply additional evidence, we also aligned reads to a “wrong” assembly, by aligning 129S1 reads to the BL6J loci). The resulting coverage track presents several gaps and blunt spikes, indicating that some parts of the assembly are not well represented by the raw ONT reads.

We have now added in the methods section how this accuracy check was performed, and added the following text in the results section:

Accuracy of the de novo assemblies at the Chr4 KZFP gene cluster (as well at another double cluster on Chr12) was confirmed by inspecting the alignment of strain specific long ONT reads (>100kb) over these loci in the corresponding assembly (Supplementary Fig. 3).

ONT read alignments for the CAST reads to the CAST strain assembly also show tiling at the KZFP gene clusters, supporting that the assembly is accurate. Since strategies to address quality and accuracy of the T2T BL6J and CAST assemblies are described in detail in Francis et al. bioRxiv 2024 article, we have not added IGV snapshots of the CAST ONT reads to the CAST KZFP gene clusters in this manuscript. However, we are including those snapshots here for completeness:

Supplementary Fig. 3 - extra

g

CAST Chr4 KZFP gene cluster region

T2T_CAST_EiJ#2#chr4:141,473,822-149,611,663

h

CAST Chr12 KZFP gene double cluster region

T2T_CAST_EiJ#2#chr12:26,589,853-32,024,578

Minor points:

Figures:

figure 1b - missing red triangles in the visual legend (although they are explained in the text of the legend, but weird double standard to have since everything else is there)

We have added the red triangles also in visual legend.

figure 1c - overlap of blue and green lines makes it likely a lot of green is completely hidden by the blue, consider putting those on two separate tracks

The overlap is due to the repetitiveness of the locus, and the fact that we have retained all primary and secondary alignments to account for duplicated regions in mouse compared to rat and also to highlight the challenges in sequence aligning this locus. Dividing the alignment in two separate tracks might make the overall figure more confusing. To improve visibility, we have filtered the alignment file to only visualize alignments larger than 3kb. This cutoff still allows to appreciate the multiple alignments within the locus but reduces the overlapping links. We have updated the corresponding methods section as follows:

For mouse versus rat comparison (Fig.1c), only alignments larger than 3kb were retained to improve visibility.

figure 3a and b somewhat redundant with panel f - panel f also implies some evolutionary sequence which might or might not represent faithfully the common ancestor at each branch point, having a tree on the side might lower the risk of this potential misunderstanding.

We disagree with the first part of this comment. Figures 3a and b provide a more detailed comparison between mouse strains, displaying also the KZFP gene annotation and we have also now added color information to distinguish coding versus non-coding genes. We think that showing the KZFP gene annotation highlights the rearrangements within the KZFP gene cluster, which becomes even more obvious when comparing the changes in directionality and coding potential of the KZFP genes. Furthermore, Figure 3b provides a comparison between BL6J and CAST strains that is not present in panel f.

We have added a phylogenetic tree (same tree as in Fig.1d) to Figure 3f.

figure 3c 55 protein-coding kzfps in c57bl6, and the sum in figure 3d counts 38 distinct arrays - however, figure 6 and the related text says 'all KZFPs in the cluster' (line 341) yet the figure 6 panel A contains only 50.

We understand that the different numbers could lead to some confusion. We annotated 55 protein coding genes in BL6J; several of them shared the same fingerprint array (as shown in Supplementary Table 2), reducing the number of distinct arrays to 38. For the functional validation of KZFPs by ChIP-seq, we have performed experiments for all coding KZFPs with at least one amino acid difference, independently of the position of the amino acid change – this means that even if some KZFPs have the same fingerprint array, we have performed ChIP-seq experiments if they had at least 1 amino acid that was different. Of the 55 protein-coding KZFP genes in BL6J, 5 of them had 100% amino acid sequence identity with another KZFP, and for these cases we only performed the ChIP-seq experiment for one of the two identical ORFs. We have included this information in the main text:

“Several fingerprint arrays were shared across multiple KZFPs and we identified 38, 47 and 45 distinct Chr4 KZFP fingerprint arrays in the BL6J, 129S1 and CAST strains, respectively.”

“...we characterized the DNA binding properties of all the KZFPs encoded in the BL6J Chr4 cluster with at least one amino acid difference – combining new ChIP-seq data for 42 KZFPs (including KZFPs with new or updated annotation) with previously published ChIP-seq data for 8 KZFPs...”

figure 3 panel d - definition of 'different' and 'distinct' might change these results quite a lot - line 147

states 'extremely disparate' - a panel in figure 3 showing the full clustering analysis of znf signatures displayed by groups would enhance the manuscript. Also using an exact match as a high threshold and doesn't imply functionally unique as it is unclear how many zinc finger domains participate in the binding for each of the identified kzfps - the variability might not be functionally meaningful (yet) - a clustered version of signatures in supplementary table 2 in some cartoon format would be nice to see.

We have changed “different” to “distinct” on top of Figure 3d. We agree that an exact match is a high threshold and doesn't imply functionally unique arrays. In Supplementary Table 6, we do provide some information about which fingers are likely to contribute to the DNA binding (analysis by the RCADE tool), and in that table we also sorted the fingerprint arrays by similarity to facilitate functional comparison (same order as in Fig. 6a).

For Fig.3d we used an exact fingerprint match as cutoff because we know from different studies that even a single substitution of a fingerprint amino acid in a critical zinc finger can modify the binding specificity of the whole KZFP array. Also, array similarities for the zinc fingers engaging in the DNA binding do not necessarily imply that the DNA binding specificity is maintained when more fingers are added. An example of such scenario is given by Zfp600 versus Rex2 and similar KZFPs: while Zfp600 shares all the fingers that in Rex2 could explain the DNA binding specificity (fingers 2-8, also shared with Zfp6003, Zfp_MB035, Zfp6001 and Zfp6002), the other zinc fingers affect the final DNA binding preference (MMETn versus RLTR4_MM-int) or lack of overall strong binding (in the case of Zfp600, which presents the largest array).

Figure 3 panel e has very limited usefulness, is unreadable unless zoomed in at 400% and would be better sent to the supplementary figures.

The graphs in Fig. 3e aim to provide an overview of the overall patterns of self-identical sequences. They are meant to be looked at the 100% magnification and the patterns are still visible even at 50% magnification. For example, it's possible to observe the higher content of self-identical sequences at the center of the CAST cluster compared to BL6J and 129S1. Similar plots have been used in published research articles to depict the repetitiveness of centromeric regions, often at a much lower zoom than our plots.

figure 4 - it would be nice to have each track be exclusive for a segmental duplication unit - the second line seems to contain two different type of units. Also, do you get the same results here comparing zinc fingerprints with a threshold of similarity (>70, >80%)

We have separated the segmental duplication units on individual tracks. It is to be noted, however, that small chunks of gene couples are present in multiple larger chunks, often with the “gain” of additional genes in between (or perhaps loss).

We did not perform this analysis based on similarity of the fingerprint amino acids, but based on the whole 3' exon sequence (including the 3'UTR). This reduced biases caused by the acquisition of mutations that could strongly affect the coding sequence and allowed a better comparison of the DNA sequence only, independently of the coding potential. The overall comparison reflects quite well the duplications, as highlighted also by the green dots in Supplementary Fig.11a, where all the genes from the

“green branch” (tree in Fig. 4a) are clearly duplications of each other. The method for the comparison is also stated in the text:

“To investigate this, we compared the sequences of the 3' exon - which encodes the zinc finger array - of all KZFP genes within the BL6J cluster, regardless of their coding potential, to identify similarity relationships between them (Fig. 4a)”.

We provide an extensive explanation about the strategy used (whole 3' exon versus fingerprint array) in reply to the comment to line 179.

Text:

It would be appropriate to say in the early parts of the manuscript that similarly large clusters of young KZFPs have not been found in most species so far. Mention of the human T2T genome (and potentially a reanalysis to search for new KZFP clusters?) would enrich the manuscript.

We agree in part. An extensive analysis accounting for the position of KZFP genes in the genome of different species is actually still missing. Most comparative studies thus far have focused on identifying conserved versus species-specific KZFP genes, independently of their location on chromosomes or of the structural architecture of the genomic clusters that harbor them. This means that we actually don't know whether similarly large KZFP gene clusters are truly absent in other species, considering that several reference genome assemblies are only available as scaffolds and still present gaps that might hide similarly large clusters in other species.

In the human genome there is an example of a ~4.5Mb large KZFP gene cluster on Chr19 (chr19: 19,668,106-24,135,000 in the most recent hg38 assembly), consisting of primate-specific KZFPs. While an in-depth analysis of KZFP gene clusters across T2T assemblies in the primate lineage is the focus of a different ongoing study, we have now included one more supplementary figure (Supplementary Fig.13) to show the distribution of human KZFP gene clusters (based on the current reference assembly hg38.p14). We provide also a TE infiltration/enrichment analysis for this locus and for another primate-specific cluster compared to the two conserved clusters examined also in rat and mice.

line 32: "Most KZFPs repress endogenous retroviruses (ERVs) and other retrotransposons"

As we don't have any data in other species, it would be more appropriate to say "Most KZFPs in human and mice have been found to repress endogenous retroviruses"

We have changed this as suggested.

line 34: "How new KZFPs emerge in response to ERV invasions is currently unknown."

"in response" is leading, "in correlation" would be more appropriate, so far there has been no great example / proof of the ERV invasion preceding the appearance of a KZFP that can recognize it. Also see line 67, 70, among others

We changed this sentence as follows. “**Whether new KZFPs emerge in response to ERV invasions is currently unknown.**” As we provide evidence of ERV infiltration prior to segmental duplications and of chimeric ERVs at some recombination boundaries, we have established that some ERV integrations happened prior to the expansion of the KZFP repertoire in young clusters.

line 39: "Our data supports a model by which new ERV integrations within young KZFP gene clusters likely promoted recombination events leading to the emergence of new KZFPs that repress them."
The evidence presented here is weak and correlative - such a strong statement in the abstract is misleading.

In our revised manuscript, we provide additional compelling evidence that at least some ERVs integrated at KZFP gene clusters prior to duplication events (new Supplementary Fig. 11). We provide additional evidence of non-allelic homologous recombination events between different ERVs (which also others have described, just not at KZFP gene clusters), which are present at identifiable recombination boundaries (new Supplementary Fig. 11). We also show that new KZFPs in the mouse Chr4 cluster can bind to mouse specific ERVs. We have provided evidence for each point mentioned in our abstract and model (please, see also our response to the point "Line 320", about the model in Figure 5b).

line 57: "While a few evolutionary old and conserved KZFPs have been shown to play essential roles in core developmental processes like embryonic development [3], imprinting [4, 5] and meiotic hotspot determination [6, 7], the vast majority of KZFPs bind to and repress transposable elements (TEs) [1]."
This sentence can give to new readers the false impression that evolutionary old and conserved KZFPs do not bind and repress TEs, which is incorrect.

We changed as follows, "While a few evolutionary old and conserved KZFPs have been shown to play essential roles in core developmental processes like embryonic development [3], imprinting [4, 5] and meiotic hotspot determination [6, 7], the vast majority of KZFPs, both young and old, bind to and repress transposable elements (TEs)"

early introduction: there is currently no mentions of the arms race scenario hypothesis derived from the ZNF91/ZNF93 data/paper.

We have added a reference to the ZNF91/ZNF93 paper by Jacobs et al. in the discussion section.

line 90 and throughout the manuscript - more precision of which assembly is discussed at any given point would enhance understanding of the manuscript.

We have stated in the figure descriptions which publicly available assembly builds were used in each figure panel, and we use consistent nomenclature for the de novo assemblies throughout the paper.

line 98 - can you confirm that both assemblies of these genomes are high quality enough that the cluster wouldn't be missed?

Yes, we have extensively looked for genome assemblies that did not have any gap at the Tnfrsf8-Miip locus, to exclude the possibility that the KZFP gene cluster was just hidden in assembly gaps. This became clear when we saw that the whole KZFP cluster locus was a huge gap in the available reference assembly of *Mus pahari* (Fig.1d). So we made sure to only retain in our analysis assemblies without gaps at the analyzed locus: the Mongolian gerbil (Bangor_MerUng_6.1) and golden hamster (BCM_Maur_2.0)

assemblies have no gaps at the *Tnfrsf8*-*Miip* locus, so we are confident that the KZFP gene cluster is absent in these species.

line 132 - extended figure 3 to me looks identical assemblies, is 'largely comparable' the most appropriate here? To me it is a weaker qualifier than it could be

We had observed the presence of SNPs when we manually inspected the genome alignments, but not structural variants, so we used “largely comparable” to account for small mutations. We have now included the SNP analysis at the bottom of Supplementary Fig. 4 (panels d and e) and have rephrased “largely comparable” to “structurally identical”.

Line 133 - "This suggests that the large differences in KZFP gene cluster size observed between the three mouse strains are due to strain specific, rather than individual, locus divergence." - although I agree this is likely, this claim would be stronger if the authors could confirm that the DNA used for the new assemblies is from multiple individual mice.

The BL6J and 129S1 assemblies in this study are the result of sequencing 2 different individuals, since the PacBio HiFi data was generated from kidney of a male mouse from each strain, while the ONT data was generated from F1 mESCs derived from breeding completely independent mice. If the 129S1 clusters were structurally divergent, it would not have been possible for Verkko to generate a gapless assembly of those loci, considering that the PacBio HiFi reads were not enough to generate the assembly (for example, the 129S1 Chr4 cluster was split into 4 contigs based on PacBio HiFi reads only).

Since the BL6J assembly generated in this study (together with the 129S1 assembly) has been a completely independent effort from the C57BL/6J T2T assembly generated in Francis et al. (together with the CAST assembly) the fact that two completely independent BL6J mice gave overall similar sequences (except for some SNPs and small insertions/deletions), suggests that it is very unlikely that the major cluster re-arrangements observed in the 129S1 strain compared to BL6J, or in the CAST strain compared to BL6J, are due to individual mouse differences.

line 179 - why use the full 3' end exon to calculate similarity here, as the zinc finger signature is usually more informative? Is there a reason to switch which metric is used in different parts of the manuscript? Can you provide a comparison between both and use either a single one, or some merge of both?

While fingerprint similarity is the best strategy to identify functional orthologs across species, this strategy was not applicable in the analysis shown in Fig.4.

At least for the KZFPs in the Chr4 cluster, a proper sequence similarity analysis based on the fingerprint amino acids alone would be flawed: many of the KZFP genes analyzed in this cluster retain sequences corresponding to several zinc fingers after the stop codon, and sometimes alternating between different frames in large stretches of the 3'UTR. This suggests that point mutations leading to the acquisition of a “premature” stop codon or small insertions/deletions that caused frameshifts, would compromise any comparative analysis based on the coding sequence only. Furthermore, we could compare genes independently of their coding potential, considering that some of them are pseudogenes. This is the main reason why we have used the whole 3'exon sequence (including the 3'UTR) to address gene similarity in the analysis shown in figure 4a.

We have added this explanation in the main text, and also provide a new Supplementary Fig. 5 illustrating one striking example of why comparing the whole 3' exon is better in this case instead of limiting the analysis to only the fingerprint amino acids in the coding region.

To investigate this, we compared the sequences of the whole 3' exon (including both the portion encoding for the zinc finger array and the 3'UTR) of all KZFP genes within the BL6J cluster. This strategy enabled us to compare the underlying DNA sequence of the exon that contributes the most to the individual KZFP gene identity, while disregarding their actual coding potential. Limiting our analysis to the coding region would have biased the comparison, as truncated arrays caused by isolated point mutations would appear highly dissimilar, even though the sequences are nearly identical (Supplementary Fig. 5). This analysis allowed us to identify similarity relationships between all the KZFP genes within the BL6J Chr4 cluster (Fig. 4a).

We have focused on fingerprint similarity to facilitate the comparison of the ChIP-seq data, for which only the coding sequence is informative, while comparison of the untranslated region would be meaningless to address functional similarities.

Line 301 - are those DNA transposons still active in these genomes? Here the language implies that the only way DNA transposons can increase in copy number is by segmental duplication, which is incorrect.

Most DNA transposons in mammals are inactive, with only some species showing recent DNA transposon activity, such as the *Myotis lucifugus* bat (Pace and Feschotte, *Genome Res.* 2007; Ray, Feschotte et al, *Genome Res* 2008; Gagnier, Belancio and Mager, *Mobile DNA* 2019). Also, as DNA transposons use a cut and paste mechanism, they are less efficient at increasing their copy number compared to active retrotransposons.

Line 320 'support the following model', 'suggest' would be more appropriate

Most suggested mechanisms in this section are weakly supported by the data presented, and so is the model in figure 5b which could use a few more question marks to make it clear that these links are suggested - this is the kind of figure that along the way gets interpreted literally and clouds the understanding of newcomers to the field in the long term.

We have changed this as suggested. We have built our model based on observations described in this study, as well as upon previous studies in the field of transposable elements and KZFP genes, but made it anyway clear in the figure description that this is a proposed model. We found evidence of new ERV integration by identifying the presence of ERV families that are unique to mouse compared to rat. Independently of how many copies of those ERVs resulted from duplications, several of those ERV families did land in the cluster after diversification of rat and mouse, but prior to segmental duplications. The large segmental duplications encompassing several genes (comparison of Fig.4c and Fig.4d makes it obvious that the gene blocks we found as duplicated are indeed nearly identical) as well as comparison of TE patterns in Supplementary Fig.11 make it clear that several TEs got duplicated. We also found evidence of chimeric ERVs that have been shown by others to represent recombination scars between different ERVs. Since we could not prove the presence of deletions, we have added a question mark about

the cluster contraction. The cluster expansion model is basically a simplified illustration of the data shown in Figure 4 and Supplementary Fig.11, and provides a more detailed version of a model that had already been reviewed in 2017 by Ecco, Imbeault and Trono (Development 2017, box1).

What we show in our study is that the same “mechanism” of TE mediated recombination that has been previously described and reviewed for other genomic loci can also apply to the KZFP gene clusters with implications for ERV and KZFP gene coevolution. We agree our hypothesis that KZFPs may reduce the instability of the locus by repressing ERVs is more speculative. The data shown in Figure 6a does not disprove this hypothesis, but we agree that more data is required to fully support this part of the model, which we had indeed indicated with a question mark.

line 354 - "suggesting that not all the KZFPs that have emerged thus far in the BL6J Chr4 cluster have a specific function." - here it is in reference to not finding a motif - do they have peaks outside of TEs? The language implies that the only function of KZFPs is to bind TE, which is not correct - some of these KZFPs very well could have other functions than bind and repress TEs

Good question. We show in Supplementary Table 6 that the KZFPs lacking specific TE enrichment also exhibit a generally low number of peaks, and no identifiable motif, suggesting that these KZFPs lack a defined target. One striking example comes from Zfp600: despite containing the zinc fingers responsible for specific binding in other KZFPs, it does not display good binding, as if the addition of more zinc fingers is somehow destabilizing the binding capability of this KZFP, consistent with previous ChIP-seq experiments in (Wolf et al., eLife 2020). The fact that some KZFPs might not have (yet?) a target sequence is not unreasonable, considering that these genes are just the product of sequences getting scrambled and recombined and, as far as we can tell, the locus does not harbor essential genes, since mice can survive the homozygous depletion of the whole Chr4 cluster (Wolf et al., eLife 2020). Furthermore, in the study by de Tribolet-Hardy et al. published in Genome Research in 2023 (Fig.5) it appears that some KZFPs in human appeared earlier than their target TEs, suggesting that some KZFPs might have been born without an actual function but might have been retained even in absence of a target if those genes were not harmful.

line 374 - "KZFP gene cluster loci" - it would be more appropriate to say 'evolutionary recent KZFP gene cluster loci'

We have added “**evolutionary young**”.

Reviewer #2 (Remarks to the Author):

This is an interesting and well-articulated manuscript that reports important observations on KZFPs, the largest group of transcription factors encoded by higher vertebrates yet a remarkably understudied family of proteins. Previous work demonstrated that KZFPs target ERVs and other transposable elements, which explains their high degree of lineage- or even species-specificity. Here, Todd McFarlan and collaborators zoom on mice, which are known to be subjected to ongoing ERV infiltration. Consistent with the TE/KZFP coevolution model, the authors accordingly find that KZFP gene pools differ very significantly between mouse strains. Using frontline genomic technology, they dissect one particular KZFP gene cluster, unveiling interesting information not only on the heterogeneity of the KZFPs found in different mouse strains but also on its ERV content and the role that these genetic invaders likely play not only on the expansion of KZFP genes but also on their regulation, with KZFPs in return controlling these TEs and remodeling of the cluster resulting in alterations or ERVs copy numbers.

The work is very well conducted and provides important insights in a family of proteins, the biological importance of which stands to be increasingly recognized as researchers stop being deterred by the repetitiveness of transposable elements-embedded sequences and being misled by the unfounded belief that KZFPs are largely redundant. Still, and no less significantly, the present study gives an important warning about the additional degree of attention to be paid if one works in the mouse model.

I do not have significant criticisms about the approaches and the results or their discussion. However, I suggest that the authors attempt some parallels with primate KZFPs, stressing similarities and differences with rodents. For instance, they could focus on the fifty or so KZFPs that appear to be conserved between mouse and human, and comment on the organization of their respective genomic environments, and try to assess whether these have been subjected or immune to TE infiltration after they were first established.

We agree that an in-depth analysis of KZFP gene clusters in the primate lineage and its comparison to rodents is important and is therefore a major focus moving forward. While we could not reasonably complete this analysis during the review period, we have added a new supplementary figure (Supplementary Fig.13), where we demonstrate similar KZFP cluster expansions in primate's concomitant with ERV infiltration at the largest human KZFP cluster. This is in contrast to older KZFP clusters, which lack a high density of recent ERV integrations. This data supports the idea that evolutionary forces driving KZFP expansion recently in rodents may also have played an important role during Hominoidea evolution.

We have added the following text to our conclusion section:

A similar trend of ERV infiltration and KZFP gene cluster expansion is also evident in the human genome, where large primate-specific clusters are heavily infiltrated by primate-specific ERVs, in contrast to conserved KZFP gene cluster loci (Supplementary Fig.13).

Reviewer #3 (Remarks to the Author):

The authors explore the distribution of KZFP genes clusters in the entire genome of mouse and conduct comparative genomics with other rodents. They focus on a cluster on Chr4, specific to the Murinae clade, thus very recent in evolutionary terms. They generate de novo genome assemblies to fill the gaps in this locus and other young KZFP gene clusters for two widely used laboratory mouse strains C57BL/6J (BL6J) and 129S1/SvImJ (129S1). Through this process, they discover, annotate, and curate new genomic sequences. To these data, they add publicly available comparable data for CAST/EiJ (CAST). The researchers perform a comparative analysis of the KZFP genes cluster on Chr4 in the three strains and determine that it evolved under parallel evolution through duplications within the locus. For BL6J they conclude that large segmental duplications were responsible for the expansion of this locus. They determine no effect of chromosomal location on the extent of heterogeneity within the locus, contrasting the cluster on Chr4 with another two young clusters (one in a similar telomeric position and one in a centromeric position). Then they contrast these three young clusters with older clusters they found, noting an abundance of small self-identical repetitive sequences in younger compared to older clusters, and also greater divergence in younger compared to older clusters.

Next, they explore how transposable elements (TEs) influenced the evolution of KZFP and found chimeric LINE-ERV elements in these regions, suggesting that non-allelic homologous recombination involving these elements likely drove KZFP cluster expansion in mice. The authors suggest that when KZFP clusters grow through duplication events, they simultaneously accumulate more transposable elements, creating more potential recombination sites, essentially creating a positive feedback cycle where expansion facilitates further expansion. They speculate that the emergence of new KZFP genes that target these newly integrated ERVs eventually slows down this self-reinforcing system. Finally, to explore the proposed mechanism, they use ChIP-seq data to show a potential negative feedback mechanism of KZFP genes on repetitive elements.

This work represents a significant advance in our understanding of KZFP gene evolution and its relationship with transposable elements. The authors make two noteworthy contributions: first, they discover substantial novel sequence absent from reference genomes, showcasing the power of long-read sequencing technology; second, they provide compelling evidence that newly expanded KZFP genes directly repress ERV expression.

Their research delivers unprecedented resolution of young KZFP gene clusters in mice through complete assembly and annotation. By resolving previously unknown sequences—nearly doubling the Chr4 cluster size—and annotating dozens of previously uncharacterized KZFP genes, the study overcomes a critical technical limitation that has hindered understanding of these genomic regions.

The methodology is sound, data analysis rigorous, and conclusions well-supported by the evidence presented. This work builds upon earlier studies identifying KZFPs as ERV repressors while advancing understanding of KZFP diversity in variable ERV regulation and the role of repetitive sequences in structural variation. What distinguishes this paper is its comprehensive integration of long-read sequencing, detailed genomic annotation, and functional characterization. I have only a few minor comments outlined below.

- Minor Comments

+ The paper would benefit from including brief explanations of technical concepts when they're first introduced. For instance, when discussing "fingerprint arrays," a simple definition explaining that these are the specific amino acid positions within zinc fingers that determine DNA binding specificity (if I get it right!) would help readers from outside the field. Adding these concise explanations for specialized terminology would make the research more accessible to readers outside this specialized field.

We have added more details about the fingerprint amino acids:

Fingerprint amino acids - corresponding to the amino acids at the positions -1, +2, +3 and +6 within each zinc finger according to helical nomenclature - are major determinants of the KZFP DNA binding specificity as they directly contact the target nucleotide sequence. Thus, we focused on the arrays of fingerprint amino acids of the coding KZFP genes and identified the repertoire of distinct fingerprint arrays in the Chr4 cluster of each mouse strain (Supplementary Table 2).

+ It would be interesting to discuss whether the authors expect variability at the population level within the same species. While long-read data with sufficient accuracy for this analysis are not yet available at population scale, making direct evaluation currently impossible, these large rearrangements appear to be relatively old events (even the more recent ones). In my view, this suggests we should expect further divergence within species, possibly contributing to regulatory differences between individuals of the same species.

We have performed a SNP analysis based on comparison of the two independent BL6J assemblies (Supplementary Fig. 4). For the Chr4cl, we found several SNPs and small in/dels, however none of them is affecting the KZFP gene coding potential. However, we cannot exclude that expanding this analysis to more individuals and across different strains might not reveal more individual and intra-species heterogeneity. We have added one sentence about this in the discussion section:

Although we could not observe large variation of the examined KZFP gene clusters between two BL6J *Mus musculus* individuals, we cannot exclude that pangenome analyses across larger populations and in different mouse strains and species might reveal more intra-species heterogeneity at evolutionary young KZFP gene clusters.

Reviewer #4 (Remarks to the Author):
